# SEDIA: A Platform for Semantically Enriched IoT Data Integration and Development of Smart City Applications

**Dimitrios Lymperis and Christos Goumopoulos** *

Information and Communication Systems Engineering Department, University of the Aegean,
83200 Samos, Greece; dlyberis@gmail.com
* Correspondence: goumop@aegean.gr

**Abstract:** The development of smart city applications often encounters a variety of challenges. These include the need to address complex requirements such as integrating diverse data sources and incorporating geographical data that reflect the physical urban environment. Platforms designed for smart cities hold a pivotal position in materializing these applications, given that they offer a suite of high-level services, which can be repurposed by developers. Although a variety of platforms are available to aid the creation of smart city applications, most fail to couple their services with geographical data, do not offer the ability to execute semantic queries on the available data, and possess restrictions that could impede the development process. This paper introduces SEDIA, a platform for developing smart applications based on diverse data sources, including geographical information, to support a semantically enriched data model for effective data analysis and integration. It also discusses the efficacy of SEDIA in a proof-of-concept smart city application related to air quality monitoring. The platform utilizes ontology classes and properties to semantically annotate collected data, and the Neo4j graph database facilitates the recognition of patterns and relationships within the data. This research also offers empirical data demonstrating the performance evaluation of SEDIA. These contributions collectively advance our understanding of semantically enriched data integration within the realm of smart city applications.

**Keywords:** smart cities; geospatial data; Internet of Things (IoT); semantic enrichment; air pollution; Air Quality Index (AQI); Neo4j; GraphQL

## 1. Introduction

The development of IoT-driven smart applications has become essential in addressing sustainability challenges and improving the quality of life for citizens in smart cities [1]. In this context, information and communication technologies have enabled smart cities to collect and analyze vast amounts of data from diverse sources such as sensor networks, public data sources, and personal devices. The data are then used to develop applications that enhance city services, support economic development, and improve society's well-being. However, developing smart city applications presents various challenges, such as integrating heterogeneous data sources and accounting for geographical information that accurately represents the urban environment [2]. A platform that supports the integration of semantically enriched IoT data and the development of smart applications can help overcome these challenges.

Smart city platforms are critical in realizing the development of smart city applications by providing high-level services that can be reused by developers. To achieve this, it is essential to ensure interoperability and effectively manage large-scale heterogeneous data [3,4]. In addition, smart city platforms must also provide secure access to data. This involves implementing robust security protocols that protect sensitive data from unauthorized access or breaches. Ensuring data privacy and confidentiality is also important, especially when dealing with personal or sensitive data.

Furthermore, smart city platforms must effectively incorporate geographical information to enable applications that can accurately represent and address urban environments [5]. This requires integrating location-based data with other data sources to provide insights into different aspects of urban life, such as transportation, energy consumption, and environmental factors [6]. Geospatial analytics can also help in understanding the relationships between different variables in urban environments.

Lack of standardization is another challenge in the development of smart city applications. To ensure interoperability and scalability, it is important to establish common standards and protocols for data sharing and application development [7]. In this context, the use of a standardized and semantically enriched data model is critical for enabling effective data integration and analysis in smart city platforms [8]. This involves developing ontologies and taxonomies that can help to capture the meaning and relationships between different data elements. By providing semantic support, smart city platforms can help to improve the accuracy and relevance of data analysis, thereby enabling the development of more effective and targeted applications [9,10].

Recent literature reviews have identified a collection of platforms designed to facilitate the development of smart city applications [11,12]. These reviews also highlighted crucial requirements to be addressed, such as device, event, resource, and data management, data processing, external data access, application runtime support, as well as maintaining a city model and historical data. Upon examining some of the existing solutions, it becomes evident that most of them take into account important requirements, such as data collection, management, and sharing. However, they do not incorporate a diverse array of services related to geographical information, nor do they perform semantic queries on the data.

In response to the identified shortcomings, this paper presents a new platform for semantically enriched IoT data integration and the development of smart city applications, referred to as SEDIA. This platform is capable of processing heterogeneous data streams from varying sources, including geographical data, by leveraging a semantically enriched data model. This unique approach significantly simplifies data integration and facilitates enhanced data analysis. To validate our solution, we further explore a practical smart city application, revealing the efficacy and potential of SEDIA in actual, real-world situations. The contribution of our work can be summarized as follows:

i.    an architecture with components that are capable of handling diverse data sources including geographical information and support a semantically enriched data model, which can facilitate effective data integration and analysis;

ii.   an implementation of the proposed architecture on top of an existing IoT middleware by enhancing its services, capitalizing on abstraction levels, and fostering interoperability;

iii.  a discussion on the efficacy of SEDIA in a proof of concept smart city application related to air quality monitoring; and

iv.   a demonstration of experimental data in relation to a performance assessment of SEDIA.

Overall, this research is distinct in that it provides a holistic approach covering all aspects of data management, from collection and semantic annotation to storage, analysis, and presentation. The emphasis that SEDIA places on semantic enrichment and the integration of geographical relationships represents a novel focus not typically prioritized in related research. The integration of an enriched data model, practical implementation, case study, and performance assessment offers a well-rounded examination of the proposed platform, going beyond the typical scope of related studies. This multi-faceted approach, combined with the innovative aspects of SEDIA, constitutes a significant contribution to the field.

The structure of this paper is organized in the following manner. Section 2 presents the SEDIA architecture showcasing our first significant contribution. Section 3 provides a comprehensive exploration of our second and third significant contributions. In particular, it presents an implementation of the proposed architecture in terms of a proof of concept application for environmental monitoring and "Green Route" identification. Section 4 is dedicated to our fourth key contribution by presenting the results of computa-

tional experiments aimed to evaluate the performance and the scalability of the proposed architecture. Section 5 provides a concise overview of previous research on semantic environmental IoT and smart cities. Section 6 discusses a summary of the research insights and a brief exploration of potential directions for future research. Lastly, Section 7 contains concluding remarks.

## 2. SEDIA Architecture

### 2.1. Overview and Principles of SEDIA Architecture

The SEDIA architecture is designed to integrate data from heterogeneous sources, potentially located in different geographical locations, and to be utilized across a wide variety of application areas. This approach to data integration enables a more comprehensive understanding of urban environments by providing a unified view of data from different sources. The proposed architecture collects, stores, and processes data by integrating various IoT technologies into the existing communication infrastructure. A critical stage of data integration involves gathering, organizing, and preparing the data for semantic labeling. Utilizing semantic tools and methods, high-level data analysis is conducted to extract information for the implementation of new functions and services within the Application Layer. The overall SEDIA architecture, as illustrated in Figure 1, comprises multiple layers. These are briefly described in the subsequent paragraphs.

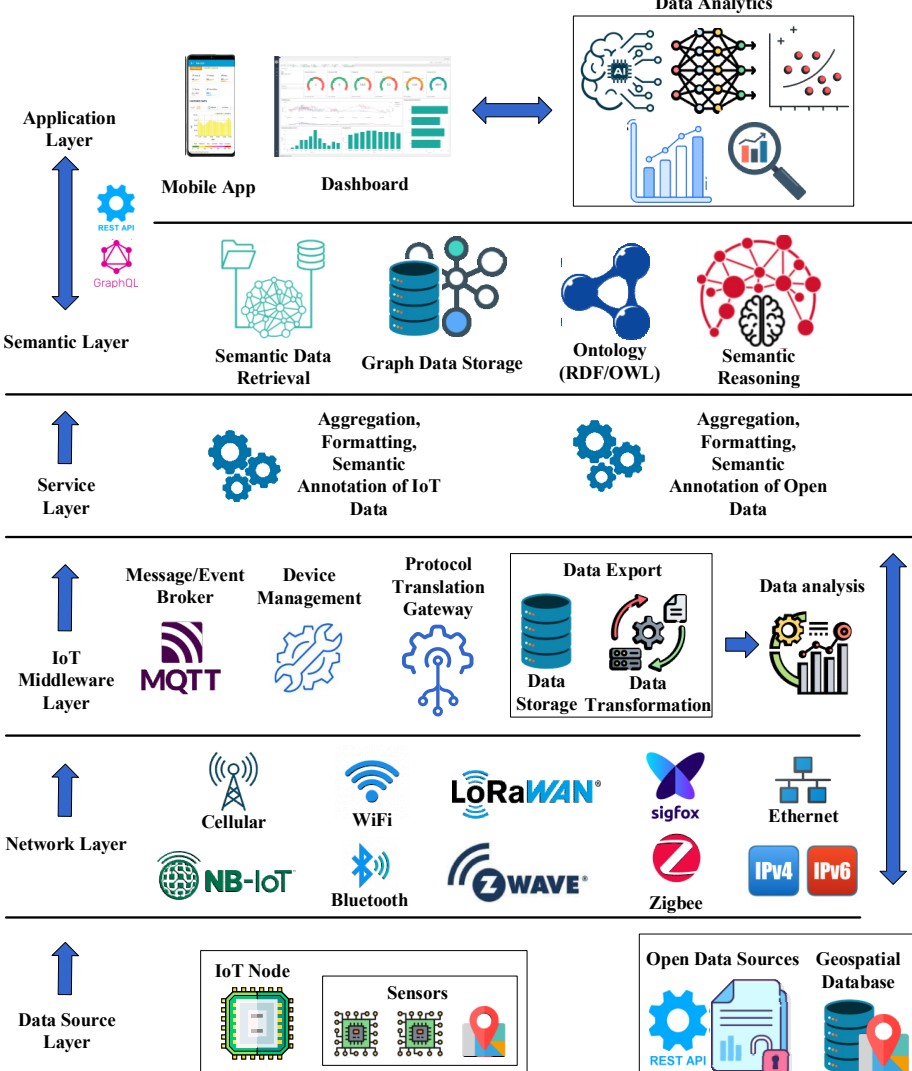

**Figure 1.** Overview of SEDIA architecture.

In an IoT system, various components can be executed at different tiers, namely the Edge, Fog, and Cloud. The placement of these components depends on several factors including specific application requirements, latency, computational resources, network bandwidth, and data privacy. The components of the SEDIA architecture can typically be assigned to these tiers as follows:

- **Edge Tier**: This tier is closest to the data source and has the least latency, but usually has limited computational resources and storage. The Data Source Layer, which includes the IoT nodes themselves and immediate data collection, is typically at the edge. Furthermore, some elements of the Network Layer, such as the initial data communication protocols (Bluetooth, Cellular, Ethernet, WiFi, LoRaWan) and part of the IoT Middleware Layer (specifically, Protocol Translation Gateways for immediate data translation) can reside at the edge.
- **Fog Tier**: This is an intermediate tier between the edge and the cloud. It has more computational resources than the edge, but less than the cloud, and it offers reduced latency compared to the cloud. More of the Network Layer (such as network gateways), as well as elements of the IoT Middleware Layer (such as Message Brokers and Device Management), can reside in the fog. This layer can also host parts of the Service Layer, especially if faster response times are needed for certain operations such as data aggregation and formatting.
- **Cloud Tier**: This tier has the highest computational power and storage capabilities, but also potentially higher latency due to its distance from the data source. Most of the IoT Middleware Layer (for more complex operations), the Service Layer (for complex data processing), the Semantic Layer (for intensive data analysis and reasoning), and the Application Layer (for user interaction and data visualization) would typically reside in the cloud.

The exact distribution of these components may vary depending on the specific needs of the IoT application. Ultimately, the decision on where to execute each part will be guided by the need to strike a balance among factors such as latency, computational resources, cost, and data privacy considerations.

### 2.2. Data Source Layer

The main purpose of this layer is to collect data from multiple sources and transmit it to upper levels for further analysis and decision-making. This layer consists of sensors, smart meters, and actuators that monitor the operation of services, activities, or equipment. Moreover, this layer contains datasets supplied by government agencies or open data sources through application programming interfaces (APIs). In an environmental application scenario, the Data Source Layer would gather data relevant to monitoring and understanding environmental conditions. This could include data from temperature, humidity, and air quality sensors, as well as from geolocation devices, in order to understand the geographical distribution of environmental data and to correlate it with specific environmental features or events.

### 2.3. Network Layer

This layer serves as the connecting bridge between the Data Source Layer and the IoT Middleware Layer. Depending on the source type, it receives data from the source layer and transmits it using various wired or wireless networking technologies such as Ethernet, 3G, 4G, WiFi, Bluetooth, LoRa, Zigbee, Z-Wave, Sigfox, NB-IoT, IPv4/IPv6, and others. The networking layer facilitates communication between all data sources and linked systems. The security of data transfers is maintained depending on the protocol employed, thereby ensuring data integrity and privacy protection for the received data. Protocols such as Zigbee, WiFi, and LoRaWAN implement encryption to protect data in transit. For instance, WiFi's WPA2 protocol uses AES (Advanced Encryption Standard) encryption, providing a high degree of security. In the case of 4G and NB-IoT, mutual authentication is conducted between the device and the network using a shared secret key. LoRaWAN uses message

integrity checks (MIC) to ensure that the data received are the same as the data sent. Zigbee uses a method called 'out-of-band' key exchange, where encryption keys are exchanged via a different channel than the main data channel, reducing the risk of interception.

### 2.4. IoT Middleware Layer

This layer is primarily responsible for (a) integrating data received from various types of connected devices; (b) processing the incoming data; and (c) delivering the processed data to various applications or services. In addition to storing time series in a database, the IoT platform layer is responsible for converting, analyzing, and managing the data received from remote devices. Services at this tier are often accessible via HTTP-based REST APIs, while implementing the necessary MQTT (Message Queuing Telemetry Transport) protocol infrastructures to transfer data to higher-layer services or applications.

In real-world networks, not all devices communicate using the same protocol. While MQTT is a widely used IoT connectivity protocol, other devices might employ different protocols, such as DNP3 (Distributed Network Protocol 3) and IEC61850, which are extensively used in the field of power and utility systems. This variety in communication protocols contributes to the complexity and heterogeneity of the IoT network and its data. If the IoT devices use protocols such as DNP3 or IEC61850, which are not typically designed for direct internet communication, a Protocol Translation Gateway (PTG) is required to convert the data into a format that the IoT middleware can understand. This might involve packaging the data into TCP/IP packets so that the gateway can transmit the translated data to the IoT middleware over the internet using MQTT or HTTP. For example, to translate DNP3 messages to MQTT messages, the steps that must be performed by a PTG include:

- Collection of the DNP3 data from the IoT device;
- Parsing the data packets according to the DNP3 protocol specifications;
- Mapping the data fields in the DNP3 packet to corresponding data fields in an MQTT message;
- Formatting the MQTT message according to the MQTT protocol specifications;
- Transmitting the MQTT message over the internet.

A PTG can be either a device or software, and its placement can vary within the SEDIA architecture depending on the specific needs and resources of the system. When the PTG is implemented as a device, it is located at the edge of the network, close to the IoT devices. This hardware-based PTG has built-in capabilities to communicate with the IoT devices using their native protocols (such as DNP3 or IEC61850), and also the capability to connect to the Internet for forwarding translated data to the IoT middleware. On the other hand, a PTG can also be implemented as software. This software-based PTG could be deployed on edge devices, fog nodes, or even in the cloud. The choice depends on various factors including network latency, computational resources, and security requirements. For example, deploying the PTG software on fog nodes can provide a balance between the low-latency and powerful data processing capabilities, which can be particularly useful when the PTG has to handle a large amount of data from numerous IoT devices.

### 2.5. Service Layer

This layer represents the apex of the data collection strategy, incorporating specialized services for real-time semantic labeling of the data. During this phase, the IoT data retrieved from the lower layer are transformed into a singular format and are prepared for performing semantic labeling based on the ontology established in the upper layer. In addition, information is collected from open data sources, which are subjected to the same procedures as IoT data.

### 2.6. Semantic Layer

At this layer, gathered and organized sensor and open data are represented utilizing semantic web technologies, such as RDF (Resource Description Framework), OWL (Web Ontology Language), and SPARQL (SPARQL Protocol and RDF Query Language), which

allow for the annotation and linking of data to facilitate more accurate and meaningful analysis. The data are converted to their semantic form, categorized according to the SEDIA ontology, and stored in graph databases. Upon storage, the semantic data retrieval process is initiated. The collected data are extracted from the graph databases using semantic queries, often formulated in SPARQL, which enable the identification and retrieval of relevant data by searching for specific patterns, relationships, and attributes. The retrieved data can then be utilized in further analysis or decision-making processes. In parallel with data retrieval, semantic reasoning takes place, involving the application of logical rules and inferences to the data stored in the graph databases. This reasoning process allows for the discovery of new relationships, validation of existing knowledge, and the detection of inconsistencies within the data. By combining the use of queries and semantic reasoning, the system can find patterns and extract knowledge, such as analyzing abnormalities or crucial events, from stored data.

### 2.7. Application Layer

This layer is responsible for providing the user with specialized services. Using REST APIs or GraphQL, which enable the retrieval and manipulation of data stored in a graph database, is one method to achieve this. These APIs facilitate seamless interaction between the Application Layer and the underlying semantic databases, promoting efficient data exchange and streamlined processing. Using visualizations, data analysis, and other tools provided by the Application Layer, this data can then be presented to the user in an understandable and actionable manner. These services are applicable to numerous IoT applications, including smart residences, smart communities, smart health, and precision agriculture.

Additionally, the Application Layer may incorporate machine learning algorithms, artificial intelligence, or other advanced analytics techniques to further process and analyze the retrieved data. This enables the generation of valuable insights, predictions, and recommendations that can enhance the overall performance and effectiveness of the IoT system.

## 3. SEDIA Implementation

### 3.1. Proof of Concept Smart Application

The smart application that is used as a running example in this paper to discuss the implementation of the SEDIA architecture focuses on improving urban air quality monitoring by utilizing IoT data and applying semantic enrichment techniques. Air pollution is a serious issue that affects the health of citizens, and the development of smart city applications can contribute to addressing this issue. According to the World Health Organization, long-term exposure to suspended indoor-outdoor particles can severely impact health and even cause death [13,14]. The European Air Quality Index (EAQI) serves as a reference to assess air pollution severity and identify the contributing factors [15]. Such an application requires services and methods for collecting large volumes of geospatial and environmental data using sensors embedded in heterogeneous IoT systems. These systems consist of nodes with embedded microcontrollers of various technologies and manufacturers, employing different communication and networking methods.

In the proof-of-concept (PoC) scenario presented in this study, an environmental monitoring methodology is followed to develop a smart application. This application is designed to provide users with the shortest route between two locations, while also ensuring the best air quality along that route. This application has been termed the 'Green Route' application. The proposed methodology consists of a data collection process from heterogeneous IoT devices and open platforms, semantic annotation of the collected data, and serving the extracted knowledge through a web application. IoT devices are strategically placed in urban environments to collect data on air quality metrics, such as concentrations of harmful pollutants ($SO_2$, $NO_2$, $O_3$) and particulate matter levels ($PM_{2.5}$, $PM_{10}$) [16]. Additionally, environmental data for particular geographical locations are gathered from open plat-

forms through APIs. This data management and enrichment process involves integrating third-party sources, often resulting in data heterogeneity and different formats. In order to overcome these challenges, the use of semantic enrichment techniques is crucial, and the semantic web can be a powerful tool in enabling the IoT to work more effectively and efficiently. Semantic enrichment enables the harmonization of data from diverse sources and formats by adding context and meaning to the raw information.

This is precisely where SEDIA comes into play. The collected data by IoT platforms are transmitted via the MQTT protocol to a central server. Custom services, written in Python, collect geospatial environmental data from open platforms, and the acquired data is processed in real time and semantically annotated by assigning ontology classes and properties. The use of semantic enrichment techniques enables a more complete understanding of the data as it adds context and meaning to the raw data collected from different sources. Additionally, storing this semantically enriched data in a Neo4j graph database facilitates the identification of patterns and relationships, making it easier to pinpoint areas of concern and potential causes of air pollution.

A web application has been developed to visualize environmental data from monitoring stations by submitting GraphQL queries to the Neo4j database. This functionality allows for the effortless identification of areas with high pollution levels and enables users to find walking routes with minimal atmospheric pollution by specifying start and end points on the map.

### 3.2. Data Source Layer

3.2.1. IoT Devices

At the Data Source Layer, SEDIA envisions collecting heterogeneous geospatial data to support smart city applications. This involves using various hardware IoT platforms, such as Arduino Uno WiFi Rev2, Arduino Mega 2560 Rev3, TTGO Lora32 V1.0, and Lopy4, which act as IoT nodes for measuring crucial parameters. These platforms provide a variety of features and capabilities, including built-in Wi-Fi and Lora networking, which facilitate data transmission over long distances with minimal power consumption. However, the use of multiple hardware platforms also presents programming and data integration challenges. Each platform may require different programming languages, such as C/C++ and MicroPython, and the use of different programming tools. This heterogeneity in hardware and programming languages necessitates careful consideration during the application implementation, including the development of a unified communication protocol and the maintenance of interoperability between different platforms. Such challenges can be addressed by adopting standardized communication protocols, data formats, and middleware solutions that enable seamless data exchange and integration across diverse devices and platforms. By fostering compatibility and simplifying the development process, these measures ultimately enhance the overall functionality, scalability, and adaptability of the smart city applications, allowing them to evolve in line with emerging technologies and requirements.

Each IoT node device employed in the PoC application is equipped with sensors for measuring critical environmental parameters, which are used to calculate the EAQI [17]. These sensors include the PMS5003 for measuring $PM_{2.5}$ and $PM_{10}$ particle matters, the low-cost MQ Gas sensors MQ131 for measuring $O_3$ and $NO_2$, and the MQ136 for measuring $SO_2$. The PMS5003 sensor measures particle concentrations based on the principle of laser scattering. The MQ131 and MQ136 sensors use a gas-sensitive film in which the electrical conductivity of the film changes upon exposure to the target gas, thereby providing a measurement of gas concentration [18]. The use of these sensors enables the accumulation of fine-grained data on air quality, which is essential for accurately determining the Air Quality Index (*AQI*) and providing real-time information to users. Figure 2 demonstrates the physical layout and interconnectivity of the sensors in the context of the IoT node, specifically designed for the PoC application.

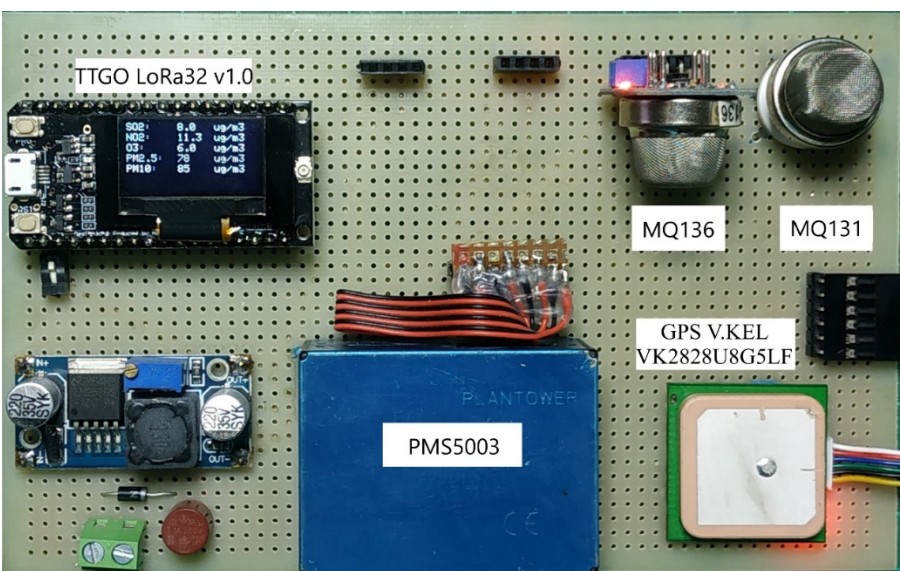

**Figure 2.** IoT prototype with peripheral sensors for the PoC application.

In addition to the sensors for measuring environmental parameters, each IoT node device in the environmental monitoring application is also equipped with a GPS device for geolocation. This GPS device acquires the altitude, latitude, and longitude coordinates of the device, which are then utilized to pinpoint the location of the collected data. This information is critical in determining the spatial distribution of air quality data and is used in generating air quality maps. By using GPS devices in each IoT node, the environmental monitoring application can collect highly accurate location data that enhances the precision of air quality measurements. This level of precision is essential in providing users with real-time air quality data that they can use to make informed decisions regarding their health and safety.

### 3.2.2. Open Data Sources

Open data sources serve as invaluable assets for promoting economic growth, stimulating innovation, and improving public services. In a smart city environment, embedded sensors within urban infrastructures and facilities possess the potential to generate a vast amount of data. APIs are widely utilized in the development of IoT solutions, enabling developers to create advanced applications that can be easily integrated with other web services.

The PoC application incorporates ambient data from open data sources, in addition to data collected from heterogeneous IoT nodes. For the application implementation, open environmental data sources with API capabilities were identified and used (Table 1). These sources provide atmospheric measurements pertaining to weather data and air pollutants for hundreds of cities around the globe, which are routinely updated and available for local use. Application developers are given unique API keys to facilitate remote call requests to these services.

**Table 1.** Open environmental data sources.

| Open Data Source | Data | Reference |
|---|---|---|
| iqair | AQI, CO, $NO_2$, $O_3$, $SO_2$, $PM_{2.5}$, $PM_{10}$ | [19] |
| ninjas_airq | AQI, CO, $NO_2$, $O_3$, $SO_2$, $PM_{2.5}$, $PM_{10}$ | [20] |
| open_weather | AQI, CO, NO, $NO_2$, $O_3$, $SO_2$, $NH_3$, $PM_{2.5}$, $PM_{10}$ | [21] |
| weatherbit | AQI, CO, $NO_2$, $O_3$, $SO_2$, $PM_{2.5}$, $PM_{10}$, pollen levels | [22] |

### 3.3. Network Layer

The Network Layer of a generic architecture supporting the development of smart city applications plays a crucial role in ensuring reliable and efficient data transmission among various devices and platforms. By employing a diverse range of networking technologies, such as Wi-Fi, Ethernet, and LoRa, as well as multiple communication protocols, including HTTP, MQTT, and LoRaWAN, this layer can accommodate the unique requirements of different IoT devices, while maintaining seamless communication.

In such an architecture, the Network Layer facilitates data transmission from IoT devices to the higher-level platforms, making use of various networking technologies and protocols that are best suited to the specific devices and application requirements. For instance, LoRa-enabled devices require a gateway that can wirelessly transmit data to the internet using the LoRaWAN protocol, thus ensuring long-range communication with minimal power consumption.

The choice of networking technologies and communication protocols is driven by factors such as the need for energy efficiency, extended communication range, and low latency. By incorporating a combination of these technologies and protocols, the Network Layer can provide a versatile and adaptable communication infrastructure, which can cater to a wide array of smart city applications.

Specifically, the IoT devices used in the PoC application leveraged a range of networking technologies and protocols to transmit the collected data to the IoT platforms in the subsequent layer. The Lorix One WiFX gateway, along with The Things Network infrastructure, facilitated efficient communication between IoT devices and platforms that support LoRaWAN networking. Despite the fact that LoRaWAN is not inherently designed for heavy data traffic or complex workload management, multiple strategies can be implemented to effectively manage workloads and handle peak traffic situations. In this study, mechanisms such as the Adaptive Data Rate (ADR) and task scheduling have been employed to ensure efficient communication under extreme conditions.

ADR dynamically adjusts the data rate and transmission power of devices, enhancing their capacity to manage data traffic while balancing range, capacity, and power consumption through the modulation of Spreading Factors (SFs); additionally, data reduction tactics can further optimize workload management. For instance, sensors located in areas with strong network coverage and minimal interference can use lower SFs, thereby achieving higher data rates and reducing transmission time, which in turn can save power and extend the device's battery life. Conversely, sensors in areas with weaker coverage or higher levels of interference may require higher SFs to ensure reliable transmission, even though this comes at the cost of reduced data rate. By modulating data rates according to each sensor's individual needs and the overall network capacity, ADR can help balance the load across the network and prevent congestion, ensuring that all collected data is reliably and efficiently transmitted to the central server for further processing and analysis.

On the other hand, by employing task scheduling, devices can be programmed to perform duties at predetermined times in order to prevent simultaneous data transmission. Scheduling can also be used to dictate when IoT devices collect data. For instance, certain environmental parameters might need to be monitored more frequently during certain times of the day or in specific weather conditions. By scheduling data collection tasks, it is ensured that these devices are actively collecting and transmitting data when they are most valuable. In environmental monitoring, some data might be more critical than others. For example, data indicating a high level of a harmful gas might be more important than data showing a slight increase in temperature. Task scheduling can ensure that critical data are collected and transmitted more frequently. In a larger-scale implementation, task scheduling can be used to evenly distribute the workload among multiple gateways and network servers, avoiding potential bottlenecks and ensuring efficient data handling.

The implemented smart application showcases the feasibility of employing diverse networking technologies and protocols in the creation of an IoT network, capable of

supporting the collection and processing of heterogeneous data for applications such as environmental monitoring.

*3.4. IoT Middleware Layer*

This layer primarily focuses on storing and analyzing data acquired from various types of connected devices. This data is generally accessible and can be retrieved through RESTful HTTP APIs or the MQTT protocol, before being relayed to higher-layer services and applications. IoT middleware solutions such as Thingspeak and Ubidots implement the MQTT protocol and possess the necessary infrastructure for receiving and transmitting data.

The MQTT Broker serves as the central communication hub, responsible for transferring messages between senders and valid recipients. Clients are connected devices, services, and applications that can either publish or subscribe to topics in order to access information. A topic contains the Broker's routing information. Clients looking to transmit messages subscribe to specific topics, while those wishing to receive messages also subscribe to particular topics. The Broker then delivers all messages containing the corresponding topic to the relevant clients. Consequently, the connection between the client and the Broker remains open, but data are only transmitted when necessary. This approach conserves battery life and network bandwidth while enhancing the real-time experience.

In the PoC application, the MQTT protocol offered by the IoT middleware was utilized for communication. This addressed the heterogeneity of networking and the use of various protocols for the transmission of data from IoT devices [23]. Information is gathered and sent to recipients with a particular information configuration model (JSON object), thereby enhancing the interoperability of the systems. The emphasis on information interoperability and the flexibility of IoT platforms make the MQTT protocol the optimal solution for interfacing with the SEDIA architecture's upper layers.

During the development of the PoC application, nine distinct IoT devices were utilized. Five of them transmit their environmental and geospatial data to the Thingspeak IoT middleware while the other four transmit their data to the Ubidots IoT middleware. The use of multiple IoT middleware solutions allows for greater flexibility in data processing and analysis [24]. Figure 3 depicts the general interconnection of the devices to the IoT middleware. The Thingspeak platform incorporates publish and subscribe functionality with a QoS level of 0 ("at most once delivery"), whereas the Ubidots platform supports QoS levels as high as 1 ("at least once delivery"). On both solutions, QoS was set to zero for faster, but less reliable, communication.

The IoT platforms employed provide access to their APIs via plain MQTT or secure MQTT with TLS (Transport Layer Security) using the endpoints presented in Table 2, depending on the type of account selected at any given moment. When implementing the interconnection of publishers and subscribers with their respective Brokers of IoT middleware, the MQTT protocol with TLS was adopted to ensure data encryption and prevent the exposure of the token API and sensor data to unauthorized parties. Certificates can be installed on programmable IoT devices and are posted on the IoT middleware websites.

**Table 2.** MQTT endpoints on IoT middleware platforms.

| IoT Middleware Platform | Connection | Broker Address | Port | Encryption |
|---|---|---|---|---|
| Thingspeak | TCP | mqtt.thingspeak.com | 1883 | - |
| Thingspeak | TCP | mqtt.thingspeak.com | 8883 | TLS |
| Ubidots | TCP | industrial.api.ubidots.com | 1883 | - |
| Ubidots | TCP | industrial.api.ubidots.com | 8883 | TLS |

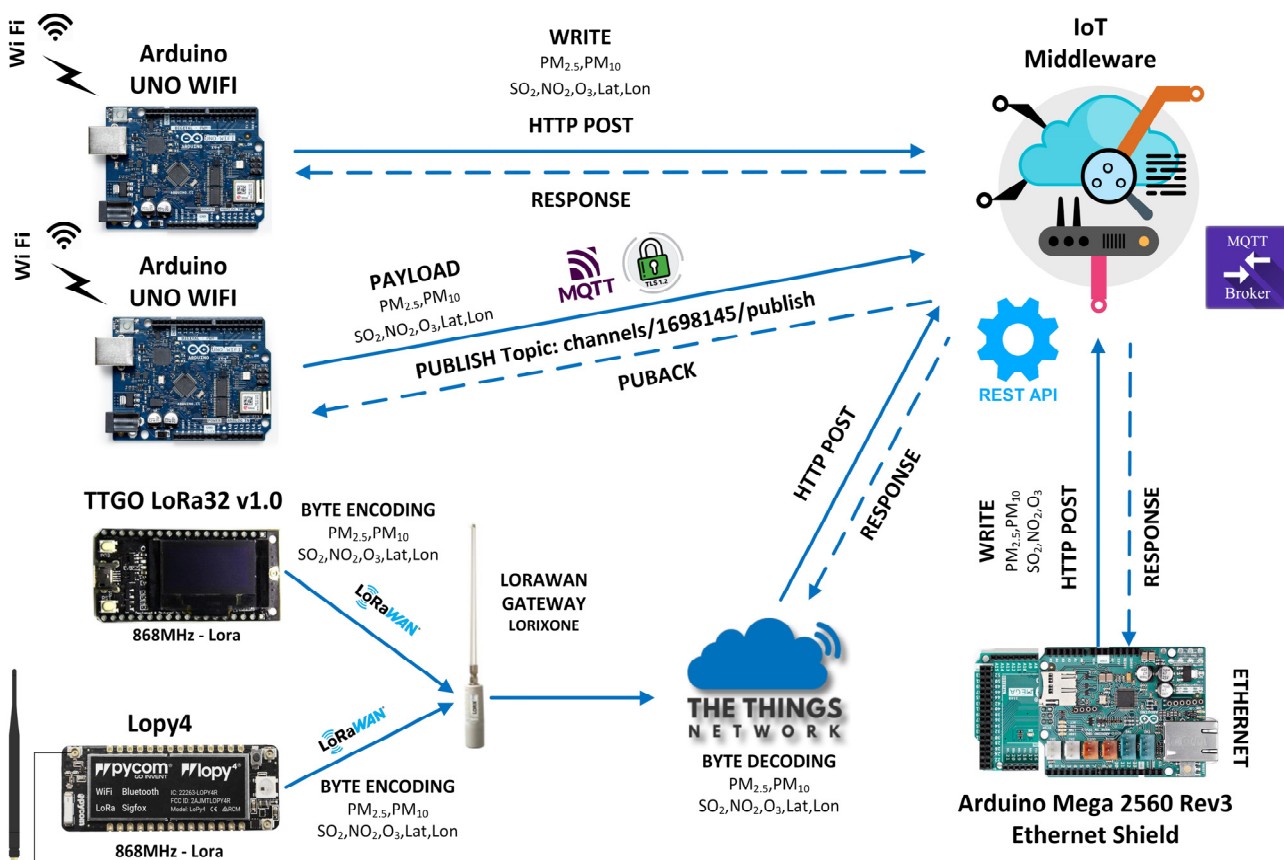

**Figure 3.** Generalized IoT device interconnection with IoT middleware platforms.

### 3.5. Service Layer

The services tier serves as the final stage in the data integration process, offering custom-built solutions for real-time semantic labeling of the data. This tier is responsible for consolidating the data collected from the IoT Middleware Layer, transforming it into a unified format, and preparing it for semantic annotation based on the ontology defined in the upper layer. Additionally, the Service Layer retrieves information from open data sources, subjecting them to similar processes as the IoT data. These custom Python services not only collect raw data from various sources in the Data Source Layer of the architecture but also filter and format the useful information, streamlining it for semantic annotation. This comprehensive approach ensures seamless integration and analysis of the collected data, paving the way for effective and efficient smart city applications. In the subsequent subsections, the methodology employed for implementing the Service Layer is discussed.

3.5.1. Multi-Client MQTT Integration: A Unified Approach for Integrating Real-Time IoT Data

The multi-client MQTT service is a custom service running on a server, written in Python programming language. Its primary functions involve gathering data sent by IoT devices to the Thingspeak and Ubidots platforms, filtering relevant information, and structuring it for improved manageability by higher levels of the architecture. Data collection from IoT middleware occurs in real time by registering the client program in all topics of the IoT devices, utilizing the MQTT protocol. The IoT data collection service can accommodate multiple clients simultaneously subscribing to topics via diverse communication channels (Figure 4).

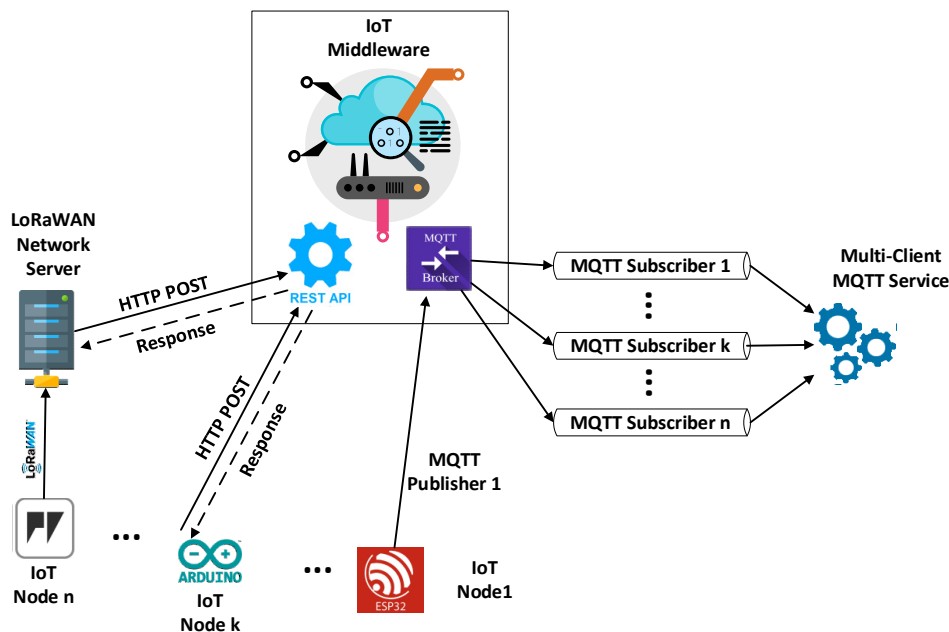

**Figure 4.** Multi-client MQTT service for IoT data aggregation.

*Eclipse Paho*, an open-source project providing high-quality implementations of M2M and IoT communication tools and libraries, is utilized for implementing subscriptions in the topics of IoT devices on each platform. *Paho Python Client* offers a client class with support for MQTT in Python, along with utility functions for easy individual message publication to an MQTT server.

The Service Layer module is composed of individual files containing function implementations that collaborate to execute the aforementioned tasks. The configuration file contains essential information about IoT platforms, data tags, and credentials for establishing connections with the respective Brokers. Upon launching the service's primary program, the configuration file's list of clients is read. Then, asynchronous connection queries are sent to MQTT servers based on the Broker's contact information and the credentials of the corresponding client. The service utilized the MQTT protocol v3.1.1, with SSL/TLS 1.3 for secure message transmission and reception. The service is designed to manage multiple client connections to Brokers using a thread pool that efficiently manages server computing resources. Additionally, the service program includes an event logging system to help understand its flow and rectify production-related issues. SEDIA generates a log file which stores information about connections to Brokers, entries in device topics on IoT platforms, and inbound messages before and after processing. This facilitates error resolution and performance analysis of applications.

3.5.2. Asynchronous Data Retrieval: A Custom Service for Concurrent Open Data Source Integration

The asynchronous data retrieval is a custom service that makes HTTP API requests to open data sources, filters the responses, and formats the data into a unified structure. Written in Python, it operates at predetermined 15-min intervals, similar to the frequency of data transmission by IoT devices. It uses asynchronous operation to simultaneously send multiple queries to various servers, ensuring smooth service operation without delays.

The service employs Python's *asyncio* library for concurrent programming, allowing multiple operations to simultaneously run using a single thread and processor component [25]. The configuration file contains connection details and geographical coordinates for querying environmental data. Upon starting the primary service program, an HTTP client session is established, and requests are placed in a task queue for asynchronous processing. Query responses are received as JSON objects and parsed to extract relevant data.

The service module includes an event logging system for debugging potential production issues. The AsyncIOScheduler runs the primary schedule every 15 min, which is suitable for asyncio-based applications as it conserves resources without requiring a new process or thread.

### 3.6. Semantic Layer

### 3.6.1. Neo4j Graph Data Storage

Neo4j is an open source graph database management system designed to store, manage, and search graph data, which consists of nodes (representing entities or objects) and edges (representing relationships or connections between nodes). Neo4j relies on the graph modeling feature, which represents data as a set of nodes and relationships that have properties (key-value pairs) attached to them [26]. This allows complex, interrelated data to be represented in a flexible and scalable way. Neo4j can be highly useful in a smart city application that involves the semantic annotation of raw data due to its inherent capabilities in handling complex, interconnected data structures. The advantages of Neo4j that make it suitable for such an application are:

- Neo4j's graph-based data representation, by effectively capturing complex interrelationships between entities such as sensors, locations, and environmental parameters, provides a more intuitive and efficient data storage and querying solution for smart city applications than traditional relational databases.
- Neo4j's scalability, capable of handling significant data volumes and traffic, ensures performance and reliability in data-intensive smart city applications by seamlessly adapting to their demands.
- Neo4j's potent Cypher query language, specifically designed for graph data, facilitates complex searches and analyses on semantically annotated smart city data, enabling efficient extraction of valuable insights and the full utilization of data potential [27–29].
- Neo4j's schema-less property graph model provides crucial flexibility for semantic annotation of raw data in smart city applications, allowing for easy adaptation to evolving data structures and relationships over time.
- Neo4j's easy integration with other semantic technologies such as RDF stores and ontology management systems facilitates a comprehensive and efficient approach to semantic annotation and analysis of environmental monitoring data.
- The robust and active Neo4j user and developer community, coupled with a rich ecosystem of tools and libraries, guarantees resource accessibility and support for developers implementing and maintaining Neo4j-based smart city applications.

### 3.6.2. Ontology

The adoption of an ontology in the proposed architecture aims to address the challenges of handling and analyzing complex, interrelated data in smart city environments. By providing a robust and flexible structure, this ontology facilitates the semantic annotation of raw data originating from diverse sources. The primary purpose of this ontology is to enable more efficient data management and processing, paving the way for advanced applications in smart cities including environmental monitoring in the form of air quality assessment, pollution detection, and decision support systems.

In this section, the ontology used to represent the data associated with the Semantic Layer of the proposed architecture is described. The ontology (Figure 5) models data from measured quantities produced by IoT devices and open data platforms. Furthermore, geospatial information from either fixed stations or measurements from mobile nodes is modeled as a potential future extension of the system. The proposed ontology, serving as a 'lightweight' version of resource and entity descriptions, contributes to reduced computing and processing time during the search phase. The ontology's simplicity renders it suitable for direct use or extension to annotate data from different application domains.

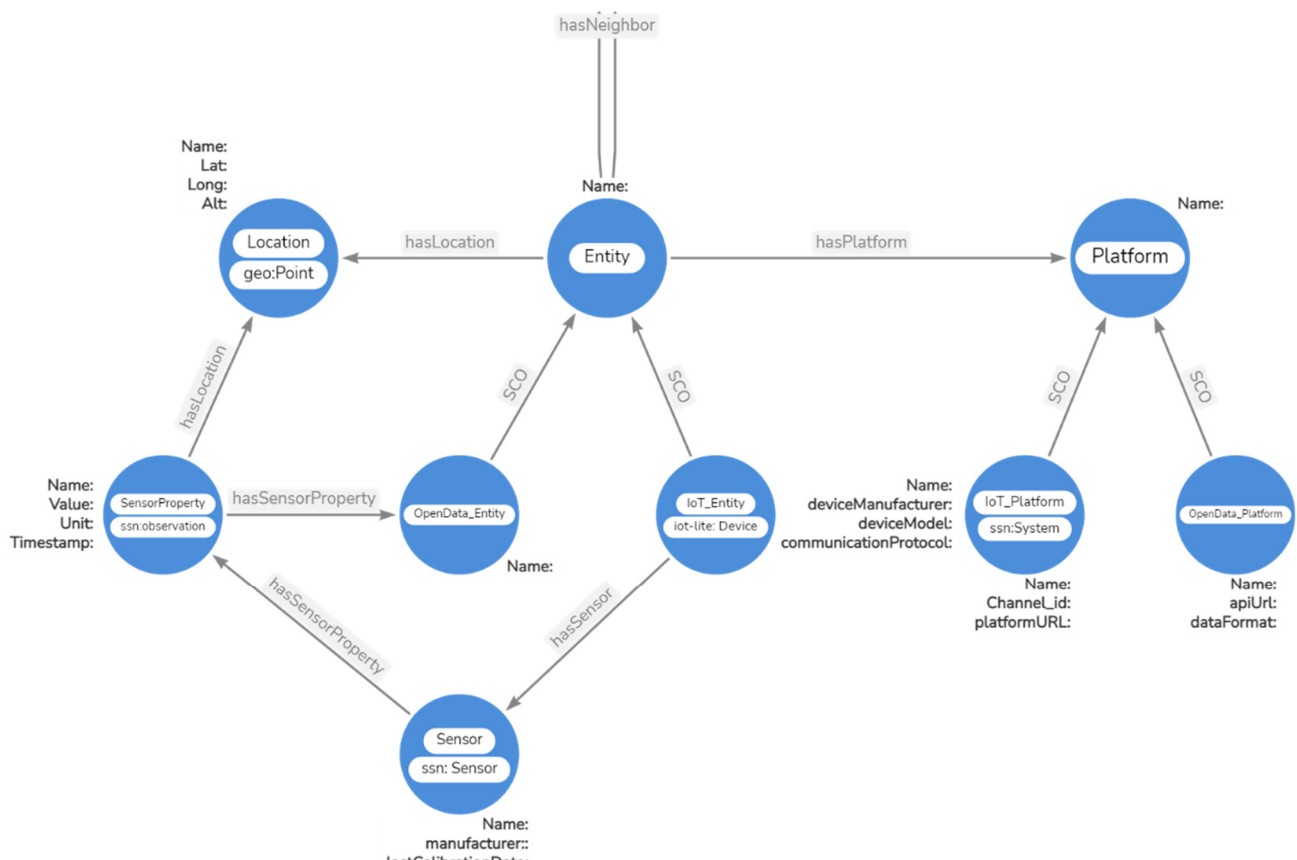

**Figure 5.** Visual representation of the SEDIA Ontology.

The ontology was initially developed in Protégé, a popular ontology editor, using the OWL format. The ontology was developed as an RDF representation to facilitate modeling complex relationships and semantic annotation of the raw data from various sources, including IoT devices and open data platforms. However, the OWL-based RDF representation of the ontology is not optimal for use in a graph database such as Neo4j. Neo4j is based on the Labeled Property Graph (LPG) model, which offers certain advantages over RDF triple stores, including simplified graph pattern matching, easier querying, and more intuitive data modeling [30].

In order to take advantage of the benefits offered by the LPG model and Neo4j, the ontology needed to be converted from the OWL-based RDF representation to an LPG representation [31]. To perform this conversion, mapping rules were established to determine how RDF classes, individuals, object properties, data properties, and annotations would correspond to LPG nodes, properties, and edges. Specifically, RDF classes and individuals were represented as LPG nodes, RDF object properties were represented as LPG edges, RDF data properties were represented as LPG node properties, and RDF annotations were represented as LPG node or edge properties, depending on the context [32].

The ontology was then exported from Protégé in an RDF serialization format. A conversion script was developed to read the exported RDF ontology and perform the transformation according to the established mapping rules. This script iterates through the RDF triples, converting classes, individuals, object properties, and data properties to their corresponding LPG elements.

The LPG representation of the ontology was loaded into a Neo4j graph database, enabling more efficient data management, processing, and querying. By converting the ontology to the LPG format and leveraging the capabilities of Neo4j, the system can effectively handle the challenges of processing and analyzing complex, interrelated data in smart city applications.

The primary ontology class, *Entity*, represents any heterogeneous data source. Two subclasses, *IoT_Entity* and *OpenData_Entity*, are derived from this class. *IoT_Entity* represents real IoT nodes, while *OpenData_Entity* represents virtual nodes with data from open sources. Moreover, the ontology facilitates the representation of platforms through the *Platform* class to describe the information's origin. *IoT_Platform* and *OpenData_Platform* subclasses are implemented for IoT platforms (e.g., Thingspeak, Ubidots) and open data platforms (e.g., iqair, open_weather, weatherbit), respectively. Table 3 summarizes the ontology classes.

**Table 3.** Ontology Classes.

| Class | Parent Class | Description |
|---|---|---|
| Entity | - | General form of system entity |
| IoT_Entity | Entity | Entities of real IoT nodes |
| OpenData_Entity | Entity | Virtual IoT node entities from open platform data |
| Platform | - | Origin of the data |
| IoT_Platform | Platform | Platforms specifically designed for managing and exchanging data from IoT nodes |
| OpenData_Platform | Platform | Platforms that provide open access to data from various sources |
| Location | - | Node geospatial location |
| Sensor | - | Node sensor |
| SensorProperty | - | Measurement Category |

LPG edges extend the class hierarchy framework of an ontology, connecting a subject and an object with a predicate to form a semantic triplet. A key feature of LPG edges is the ability to link entity instances belonging to different classes based on semantic associations between objects. The *hasNeighbor* relation, invoking neighbor relations due to the distance between two entities belonging to the *Entities* class, is particularly noteworthy. This relation is employed at the reasoning level in the PoC application to identify walking paths free of atmospheric pollutants. Table 4 summarizes the ontology relationships.

**Table 4.** Ontology Relationships.

| Relationship | Subject | Predicate | Description |
|---|---|---|---|
| hasLocation | Entity, SensorProperty | Location | Entity or SensorProperty *hasLocation* Location |
| hasPlatform | Entity | Platform | Entity *hasPlatform* Platform |
| hasSensor | IoT_Entity | Sensor | IoT_Entity *hasSensor* Sensor |
| hasSensorProperty | OpenData_Entity, Sensor | SensorProperty | OpenData_Entity or Sensor *hasSensorProperty* SensorProperty |
| hasNeighbor | Entity | Entity | Entity *hasNeighbor* Entity |

LPG node properties are specific to each class and describe the raw data received from respective sources. Expressed in various data types these properties identify the class they belong to and the type of data they describe. Table 5 summarizes the ontology node properties.

The proposed ontology reuses multiple established ontologies to model IoT and geospatial concepts, ensuring interoperability and a semantically rich representation. In particular, the ontology leverages the Semantic Sensor Network (SSN) ontology [33], IoT-Lite ontology [34,35], and GeoSPARQL ontology [36]. The SSN ontology is utilized to represent sensor-related concepts. The *Sensor* entity is aligned with the *ssn:Sensor* class, and the *SensorProperty* entity is aligned with the *ssn:Observation* class. Additionally, the *IoT_Platform* entity is aligned with the *ssn:System* class, treating an IoT platform as a system in the context of IoT. The IoT-Lite ontology is employed to represent IoT-specific concepts. The *IoT_Entity* is aligned with the *iot-lite:Device* class, enabling the integration of device-related information such as manufacturer, model, and communication protocol.

The GeoSPARQL ontology is used to represent geospatial concepts. The *Location* entity is aligned with the *geo:Point* class, providing a standard way to model geographical coordinates such as latitude, longitude, and altitude. Reusing these ontologies is primarily intended to promote interoperability with other systems and data sources employing the same ontologies. It facilitates data integration and exchange between diverse IoT and geospatial systems by ensuring the consistent and accurate representation of concepts and relationships.

**Table 5.** Ontology Node Properties.

| Property | Class | Type | Description |
|---|---|---|---|
| channel_id | IoT_Platform | Integer | Platform channel id |
| platformURL | IoT_Platform | String | The URL of the IoT platform's API |
| apiUrl | OpenData_Platform | String | The base URL of the platform's API |
| dataFormat | OpenData_Platform | String | The format in which the open data platform provides the data |
| deviceManufacturer | IoT_Entity | String | Manufacturer of the IoT device |
| deviceModel | IoT_Entity | String | The specific model of the IoT device |
| communicationProtocol | IoT_Entity | String | Communication protocol used by the IoT device to transmit data |
| manufacturer | Sensor | String | The sensor's manufacturer |
| lastCalibrationDate | Sensor | dateTime | The most recent date on which the sensor was calibrated |
| Alt | Location | Double | Altitude |
| Lat | Location | Double | Latitude |
| Long | Location | Double | Longitude |
| Unit | SensorProperty | String | Measurement unit |
| TimeStamp | SensorProperty | dateTime | Timestamp |
| Value | SensorProperty | float | Measurement value |
| Name | Entity, Platform, IoT_Entity, OpenData_Entity, Location, IoT_Platform, OpenData-Platform, Sensor, SensorProperty | String | The name of each entity |

### 3.6.3. Semantic Annotation

The semantic annotation service, developed for the purpose of semantically describing the data and recording it in the Neo4j database, is a custom service that processes data received from the services defined in the Service Layer. Semantic annotation is carried out according to the ontology, which models the structure of the data as mentioned earlier. The service is written in Python and its main functions are to receive filtered data from the Service Layer, model it, and write it in the form of nodes and relations in the Neo4j graph database. The *Neo4j Python Driver* is used for interfacing with the Neo4j graph database, enabling the creation, reading, updating, and querying of graph data.

The service program consists of two separate modules. One module implements the Neo4j base connection establishment functions, and the other implements the semantic annotation functions. The configuration file contains the base URI, password, and the maximum total number of connections allowed per host to be managed by the connection pool. The file that establishes a connection to the database according to the connection information contained in the configuration file has global driver and logger variables, so they can be accessed by other parts of the program. Additionally, it includes functions to terminate the database connection and establish a new one if necessary.

Three functions are defined that model the data of IoT nodes and open platforms according to the defined ontology. One function models the data collected by the Thingspeak platform, with function arguments being the query in Neo4j's Cypher language and the data properties for each IoT node connected to the Thingspeak platform. Another function models the data collected by the Ubidots platform and is similar to the previous function

in terms of arguments. This function checks whether the name of the variable refers to geospatial data or values of measured quantities in order to execute the corresponding Cypher code. The last function models the data collected from open data platforms, with arguments being the Cypher query and the data properties.

In Neo4j, a transaction is a series of operations performed together as a single, individual unit of work. The Python driver for Neo4j provides a way to work with transactions using the session object. This object simplifies the process of executing a Cypher statement within a database write transaction. It handles starting the transaction, committing changes, and handling any errors that may occur.

Considering the way the driver function works, two additional functions were created to handle write transactions for both IoT data and data from open platforms. One function is responsible for recording IoT data and processes the message from the IoT platforms. It establishes a session and performs the registration transaction according to the appropriate functions. Similarly, another function is responsible for writing data from open platforms and processes the message from the IoT platforms' data retrieval requests. A session is established, and the registration transaction is performed according to the relevant function. These functions are directly called from the data collection services of the Service Layer. In both cases, after the payload configuration is complete, a session is established, and data write transactions are performed.

### 3.7. Application Layer

In the proposed SEDIA architecture, the Application Layer comprises the software and services constructed on top of the foundational platform infrastructure, with the purpose of delivering distinct functions and services to users. This layer is responsible for interpreting data gathered from various sources while presenting a user-friendly interface and facilitating system interaction. The Application Layer holds critical importance within the overall architecture, as it furnishes essential features that render the system a valuable tool for its users.

Application development generally adheres to a three-tier model, encompassing a web application (frontend), an API tier (GraphQL), and a database (backend). In a full-stack application employing GraphQL, the interface layer is utilized on both the client and server sides. The frontend is tasked with managing the user interface and issuing GraphQL queries to the server, while the backend addresses GraphQL queries, performs required database operations, and delivers the requested data back to the client. This improves application performance and scalability [37,38]. Furthermore, using GraphQL on both the client and server allows for consistent and unified data interaction, regardless of the data's origin.

Specifically, for the PoC application discussed in this paper, the primary aim at the application level is to accumulate semantic data for determining the *AQI* within a designated geographical area of interest to the user. The ultimate goal is to propose a map route characterized by minimal atmospheric pollutants, based on calculated air quality indicators for each area. In the "Green Route" application, users input their starting location and destination, and the application computes and displays the recommended route, taking into account *AQI* values from stations in the vicinity of the points of interest.

### 3.7.1. Utilizing GraphQL API for Improved Application Layer Performance

This section focuses on the tools used in the application development, particularly those that enable the application to interact with the Neo4j database, accessing and presenting the stored data in an accessible and user-friendly format. Particular attention is devoted to the intricacies of Neo4j and the employment of the Cypher query language.

GraphQL, a specification for building APIs, offers an efficient and flexible alternative to RESTful APIs, representing complex nested data dependencies in modern applications [39]. With a strict type system, GraphQL describes the available data for an API. These type definitions specify the API, and the client application can request data based on these defi-

nitions, which also outline the API's entry points. GraphQL boasts numerous advantages over RESTful APIs, such as flexibility, improved developer experience, better documentation, real-time capabilities, and data integration from different systems. GraphQL uses a strong type system to specify an API's capabilities. The GraphQL Schema defines how a client can access data, the types and fields that can be queried or modified, and the organization of data sent over the Internet [40]. The Schema is designed to match the data structure in the Neo4j database as closely as possible.

In this study, the Neo4j GraphQL library is employed to leverage its powerful feature of directly embedding Cypher queries within the GraphQL schema using the *@cypher directive*. This capability allows developers to harness the robustness of the Cypher query language for conducting intricate calculations or retrieving specific data from the Neo4j database. The @cypher directive maps query results to the declared GraphQL field and necessitates the installation of the *Awesome Procedures On Cypher* (*APOC*) add-on library.

*Apollo*, a suite of tools used for implementing GraphQL on the server, client application, or in the cloud, was instrumental in the development of the application in this study. Specifically, *Apollo Server* was used to generate the GraphQL API, *Apollo Client*, a client-side JavaScript library, was employed to query the GraphQL API from the application, and *Apollo Studio's Explorer*, a tool for creating and executing GraphQL queries, was utilized for the same.

In particular, Apollo Server is employed to build GraphQL backends, providing a set of tools and libraries for constructing and operating a GraphQL server within a Node.js environment. It is specifically responsible for defining the schema, managing queries and data modifications with resolver functions, connecting to various data sources, overseeing query performance through caching and clustering, ensuring security, and facilitating real-time data updates.

On the other hand, Apollo Client offers a set of tools and libraries for interacting with a GraphQL backend and managing client-side data. With integrations for numerous frontend frameworks, such as React and Vue.js, as well as native mobile versions for iOS and Android, Apollo Client handles client data caching and can also be employed to manage local state data [41].

3.7.2. Deriving the European Air Quality Index Using GraphQL Query

Upon accessing the application's web interface, a GraphQL query is dispatched to the database to obtain the environmental data from the observation stations, encompassing the geographical area displayed on the map. The data retrieval query seeks to extract all measured quantities ($SO_2$, $NO_2$, $O_3$, $PM_{10}$, $PM_{2.5}$) and the *AQI* values of the stations, as well as their geospatial data for visualization purposes. According to the European Air Pollution Agency, a higher *AQI* value indicates increased air pollution levels [42]. Consequently, the *AQI* is calculated by considering the maximum individual *AQI* indices of environmental pollutants (Equation (1)).

$$AQI = \max(AQI_{SO2}, AQI_{NO2}, AQI_{O3}, AQI_{PM10}, AQI_{PM2.5}) \tag{1}$$

In this study, the Cypher language's syntax limitations necessitate the use of helper functions within the Neo4j base, employing the additional APOC library for direct combinatorial calculations of data. These functions take pollutant concentration measurements as input and return an integer index ranging from 0 to 5, corresponding to the verbal description of air pollution levels. Five distinct functions were developed for the monitored pollutants, and their slopes were combined into a new function that accepts a pollutant's name and returns the corresponding index value (Figure 6).

```
CALL apoc.custom.declareFunction(
  'aqi_pm10(value::NUMBER) :: INT',
  '
   RETURN CASE
   WHEN ($value >= 0 AND $value < 20)   THEN 0
   WHEN ($value >= 20 AND $value < 40)  THEN 1
   WHEN ($value >= 40 AND $value < 50)  THEN 2
   WHEN ($value >= 50 AND $value < 100) THEN 3
   WHEN ($value >= 100 AND $value < 150) THEN 4
   WHEN ($value >= 150 AND $value < 1200) THEN 5
   ELSE -1
  END
  '
);
                        (a)
```

```
CALL apoc.custom.declareFunction(
  'AQI(pollutant::STRING, value::NUMBER) :: STRING',
  '
   RETURN CASE
   WHEN $pollutant = "pm2.5" THEN custom.aqi_pm2_5($value)
   WHEN $pollutant = "pm10" THEN custom.aqi_pm10($value)
   WHEN $pollutant = "so2" THEN custom.aqi_so2($value)
   WHEN $pollutant = "no2" THEN custom.aqi_no2($value)
   WHEN $pollutant = "o3" THEN custom.aqi_o3($value)
  END
  '
);
                        (b)
```

**Figure 6.** APOC function to calculate individual *AQI* for each pollutant. (**a**) APOC custom function to calculate *AQI* for PM10. (**b**) Merged function that calculates *AQI* for every pollutant.

By leveraging the Neo4j GraphQL library's ability to incorporate custom logic into the application API, a Query-type GraphQL query was utilized, embedding the Cypher language query code. The @cypher directive is employed to integrate the Cypher query within the Schema, mapping the query results to the specified GraphQL field. This query accepts three lists as parameters, containing names of air pollutants, suspended particles, and verbal air quality indicators, and returns the environmental and geospatial data of the stations as a list (Figure 7).

```
type Query{
  Entity_AQI(SP_List: [String!]!, PART_List: [String!]!, AQI_Values: [String!]!): [aqi_points]
  @cypher(
    statement: """
    call n10s.inference.nodesLabelled('Entity', {catNameProp: 'label', catLabel: 'Resource',
    subCatRel: 'SCO' }) YIELD node
    match (node)-[:hasLocation]-(l:Location)
    with node,avg(l.lat) as lat, avg(l.long) as long
    with node, lat,long,localdatetime({timezone: 'Europe/Athens'}) as now
    match (node)-[:hasSensor*0..1]-()-[:hasSensorProperty]->(b:SensorProperty)
    where (((b.name in $SP_List) AND (LocalDateTime(b.timestamp) >= now-duration({hours:1})))
    or ((b.name in $PART_List) AND (LocalDateTime(b.timestamp) >= now-duration({hours:24}))))
    with lat,long, id(node) as id, node.name as Entity, b.name as pollutant, avg(b.value) as value
    with lat,long, Entity, pollutant, value, toInteger(custom.AQI(pollutant, value)) as aqi_idx
    with lat,long, Entity,aqi_idx, collect({pollutant:pollutant, pol_aqi:$AQI_Values[aqi_idx],
    value:round(value, 2, 'CEILING')}) as pol
    unwind pol as pollutants
    return {name:Entity, point:{lat:lat,long:long},aqi:$AQI_Values[max(aqi_idx)],
    pollutants:collect(pollutants)}
    """
  )
```

**Figure 7.** GraphQL query to retrieve environmental and geospatial data.

Utilizing the additional *Neosemantics* (*ns10*) library, the query cypher code begins by searching for *Entity*-labeled entities and those labeled as its subclasses with the *SubClassOf* (*SCO*) relation. For each returned station node, paths featuring a *hasLocation* relation and

ending at a *Location* node are explored, calculating the average longitude and latitude associated with each station.

In this study, the query utilizes the *localdatetime* function in Cypher to obtain the current time in the appropriate geographical time zone, which is then used to filter pollutant data based on the duration between the 'timestamp' property and the current time. Air pollutant data are filtered for the last hour, while particulate pollutant data are filtered for the last 24-h period. The query matches a variable-length relationship pattern that allows for any number of *hasSensor* relationships between a starting node (which could be an *IoT_Entity* or *OpenData_Entity*) and an intermediate node, followed by a single *hasSensorProperty* relationship between the intermediate node and a *SensorProperty* labeled node. The resulting *SensorProperty* nodes are filtered based on whether their 'name' property matches any of the pollutants listed in a specific list and their 'timestamp' property is within a certain time window.

The query's final result includes the station name, its geographical coordinates, the final *AQI*, and the pollutant collection (pollutant name, *AQI*, average value). Figure 8 consolidates all aspects of the query submission process to the aforementioned database and the respective technologies employed for the implementation of distinct application tiers. The diagram demonstrates the interaction between individual components, outlining the flow of a request from the client application, retrieving environmental data from the stations, submitting it to the GraphQL API, resolving data from the Neo4j database, and returning it to the client for rendering the results in an updated view of the user interface. This comprehensive process ensures seamless data retrieval and visualization, providing users with relevant and timely information on air quality and pollutant levels within the specified geographical area.

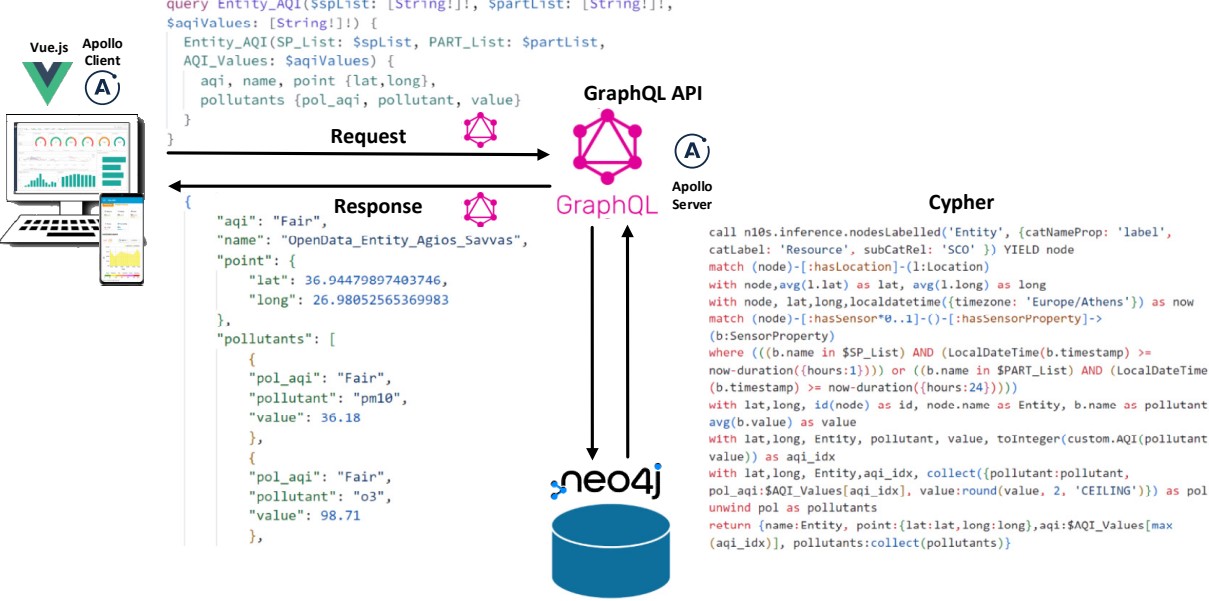

**Figure 8.** Request environmental data retrieval, through the full-stack GraphQL implementation.

### 3.7.3. Computing Shortest Paths Based on Air Pollution Data with GraphQL Mutation

After acquiring the environmental data from the stations, users are prompted to interact with the application map and input the geographical locations of their starting point and destination, enabling the application to compute the recommended route. In this aspect of the application, a GraphQL query of the mutation type was employed since it enables the writing of new data to the database. The query accepts six lists as parameters (Figure 9), which include the names of air pollutants, suspended particles, verbal air quality indicators, permissible air quality indicators, as well as the geographical starting and

ending points of the route. The query response returns a list of geospatial data points corresponding to the stations comprising the suggested route.

```
{ "spList": ["no2","so2","o3"],
  "partList": ["pm2.5","pm10"],
  "aqiValues": ["Good","Fair","Moderate","Poor","Very Poor","Extremely Poor"],
  "properAqi": ["Good","Fair","Moderate","Poor"],
  "startPoint": {"latitude": 36.964156174751494, "longitude": 26.95954263210297},
  "endPoint": {"latitude": 36.951853133035314,"longitude": 26.990141272544864}
}
```

**Figure 9.** GraphQL mutation parameters.

The route is determined by computing the *AQI* of all stations located within the broader geographical vicinity of the points of interest. The results are filtered to only retain the stations with an index corresponding to the following levels: Good, Fair, Moderate, and Poor. Following this, the query computes the kilometer distance between all potential combinations of stations, sorts these results in ascending order, and separately yields a dataset containing the four nearest nodes for each station.

Subsequently, the query creates new weighted *hasNeighbor* semantic relationships between the filtered nodes within the database; these relationships are based on the calculated distances. This process results in the creation of a graph of interconnected stations, which is subsequently used to compute the shortest "Green Route"—the series of points the user will traverse en route to their destination (Figure 10). The weight assigned to the relationships is determined by the calculated distance between the nodes [43].

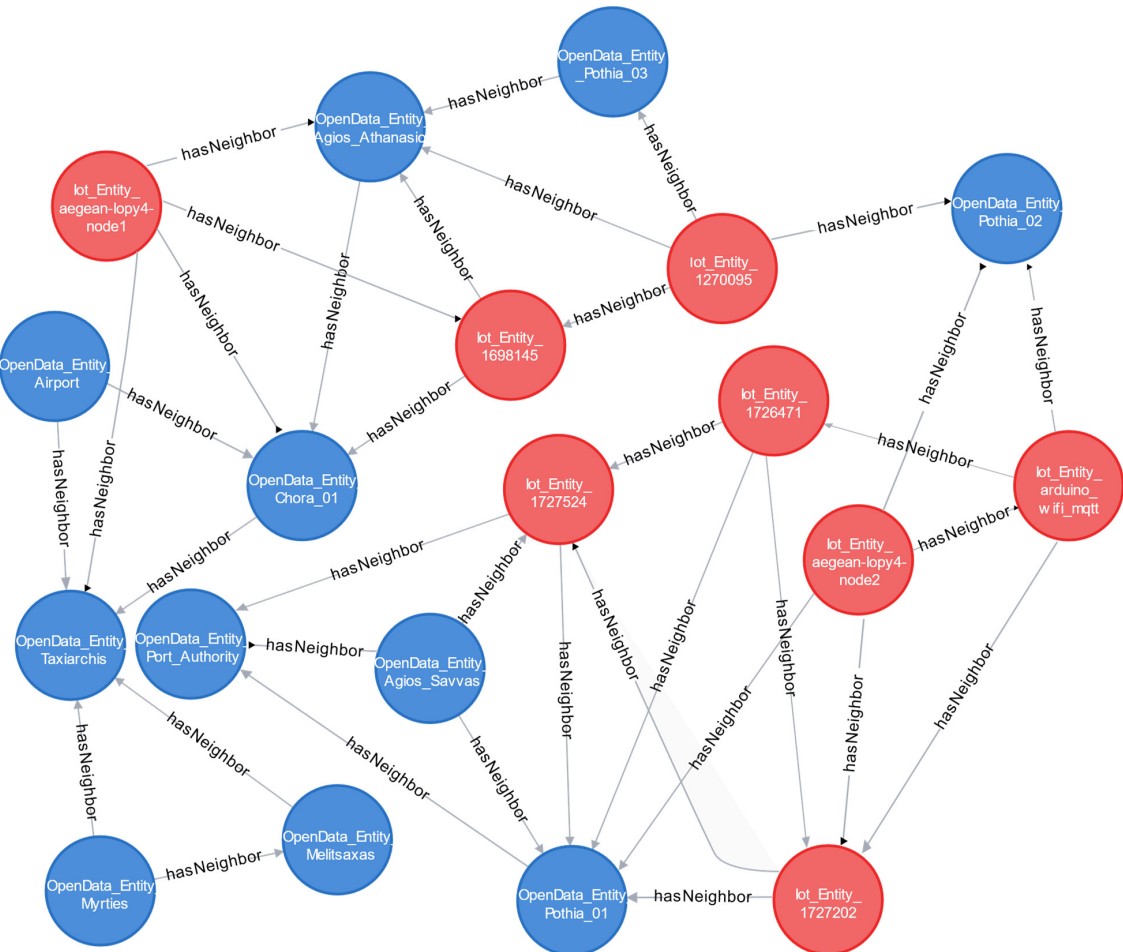

**Figure 10.** Graph of interconnected nodes with *hasNeighbor* relations.

It is crucial to highlight that the Cypher code initially incorporates the capability to delete any pre-existing *hasNeighbor* relationships between entities. This facilitates the formation of new *hasNeighbor* relationships each time the query is run. This approach guarantees the generation of new "Green Route" search graphs, reflecting the most up-to-date environmental data from the stations.

In summary, the query computes the nearest nodes to user-defined starting and ending points (start_node, end_node) within the graph. It uses Dijkstra's algorithm through the *shortestPath* function in Cypher to determine the shortest path between these nodes [44,45], which are connected by the *hasNeighbor* relation. The maximum distance allowed between the given starting and ending nodes is set to 15 hops. A succession index is computed for each node in the path, beginning with the start_node. The query ultimately returns a list of nodes in the path, comprising the name, latitude, longitude, and marker on the path, ordered by the node marker (Figure 11).

```json
{
  "data": {
    "Path": [
      {
        "idx": 1,
        "lat": 36.95664154400455,
        "long": 26.965767803949493,
        "name": "OpenData_Entity_Agios_Athanasios"
      },
      {
        "idx": 2,
        "lat": 36.964352463962754,
        "long": 26.956613291737053,
        "name": "OpenData_Entity_Taxiarchis"
      }
    ]
  }
}
```

**Figure 11.** GraphGL mutation example response.

3.7.4. "Green Route" Application Front-End

Various software technologies were combined to create a web application that presents an interactive map with routing functionality. The front-end of the developed application is a *Single Page Application* (*SPA*), a web application that dynamically updates the content of the current page through user interactions rather than loading entire new pages. This specific application consists of an HTML file, which serves as the central body of the application, while its content dynamically changes by modifying the *Document Object Model* (*DOM*). The *Vue.js* framework was employed for the visual component of the front-end application. *Bootstrap* was also used to design and build the front-end visual layout, guaranteeing dynamic content responsiveness on mobile devices.

The *Leaflet* library, an open-source JavaScript library for creating interactive web application maps, was utilized alongside Vue.js. The capabilities of Leaflet were exploited to generate customized markers, pop-ups, and other features, enhancing the user experience of the application. Furthermore, the broad compatibility of Leaflet with various mapping providers, including OpenStreetMap, Google Maps, and Mapbox, provided flexibility and variety in terms of mapping resources. Additionally, the *Leaflet-Routing-Machine* (*LRM*) plugin, which adds routing capabilities to maps, was used. This plugin is instrumental in

integrating turn-by-turn directions into the application's maps, facilitating navigation with options for various modes of transportation, waypoints, and custom markers. This plugin was employed to identify and display the suggested "Green Route" on the map.

Finally, *Mapbox*, a JavaScript library for creating vector maps, was another tool used in developing the web application. The utilization of Mapbox GL, which is constructed on WebGL, a JavaScript API designed for interactive 3D and 2D graphic rendering, offered a high-performance and highly customizable geomap rendering experience integral to the application's functionality.

Upon starting the application, a GraphQL query submission to the database is initiated to retrieve the environmental data from observation stations within the geographical area displayed on the map. The query returns all concentration values and individual air quality indicators for each pollutant monitored at each station. The overall *AQI* index of each station within the area is also provided, alongside its geospatial data for representation on the map. Stations are depicted on the map with geographical markers, the color of which corresponds to the *AQI* level (Figure 12).

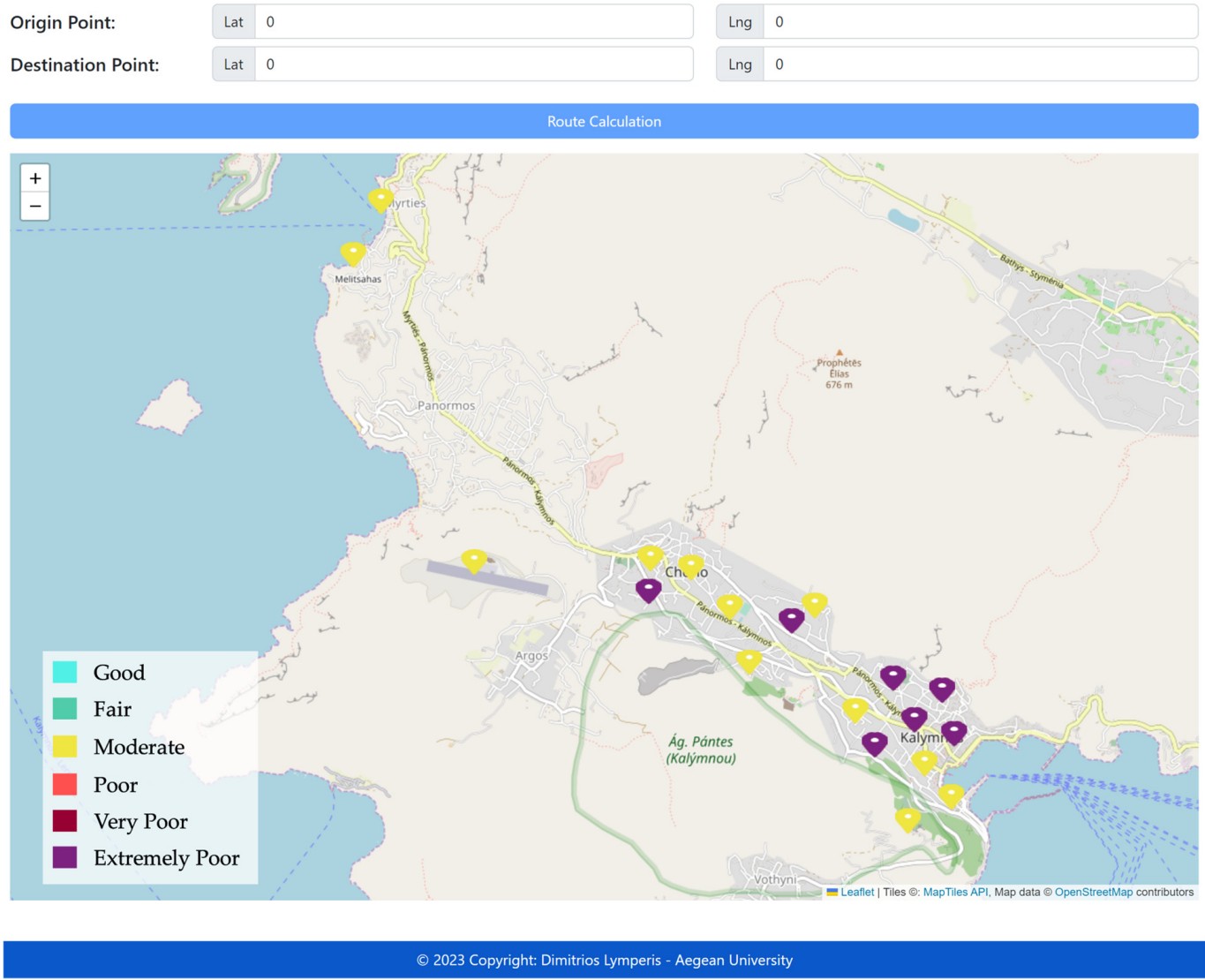

**Figure 12.** Initial application interface displaying stations returned by the GraphQL query.

Upon selecting a station's geographical marker, a pop-up window displaying the station's environmental data appears on the screen (Figure 13). The upper section of the window exhibits the overall *AQI* of the station, with its corresponding color representation.

Following this, a table presents the calculated pollutant concentrations in μg/m$^3$ and the associated *AQI* index for each pollutant. The individual coloring of each pollutant enables users to discern which pollutants contribute to the formation of the overall *AQI* index. The final section provides health-related guidelines and recommendations for both the general population and sensitive groups. These messages offer advisory information pertaining to activities that can be undertaken given the current *AQI*.

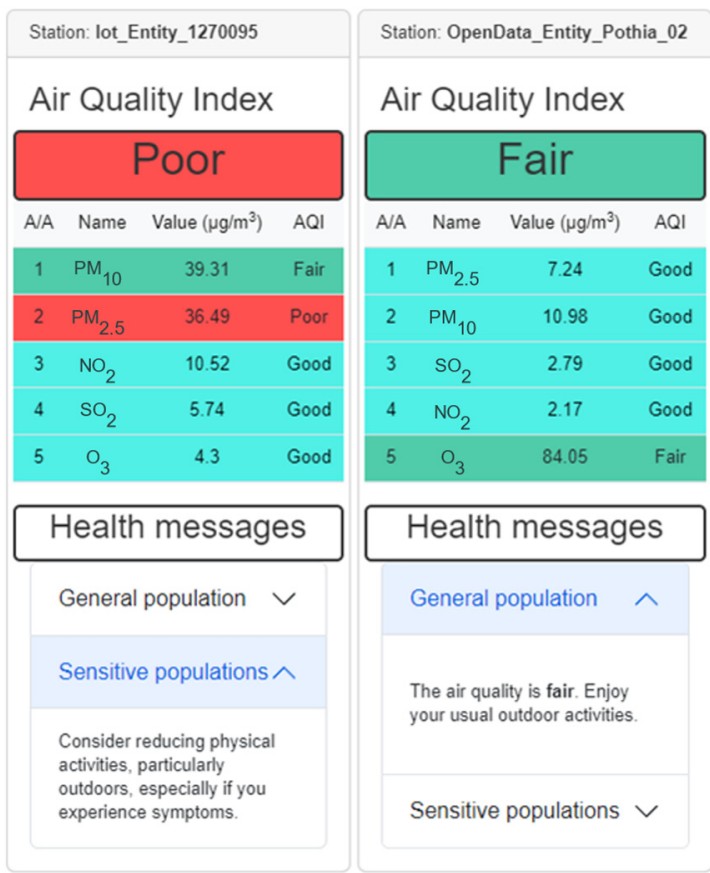

**Figure 13.** Examples of station environmental data pop-up windows.

Subsequently, the user is provided with the option to input the starting and ending points of a desired route. The selection of these points is performed by choosing the corresponding locations on the map. Upon each selection, a pop-up window emerges, prompting the user to designate whether the chosen point is the starting or the ending point. Following each selection, the geographical coordinates of the point are displayed at the top of the application. Upon clicking the "Calculate Path" button, a Mutation-type GraphQL query is submitted to the database. The query response yields a list of consecutive geospatial data for the stations, which constitute the proposed route. Subsequently, the Leaflet-Routing-Machine plugin processes the starting point, waypoints, and destination point, and computes the path. Ultimately, the application exhibits the route on the map in green, positions geographical markers in blue at the starting and ending points, and displays a window featuring route navigation instructions (Figure 14).

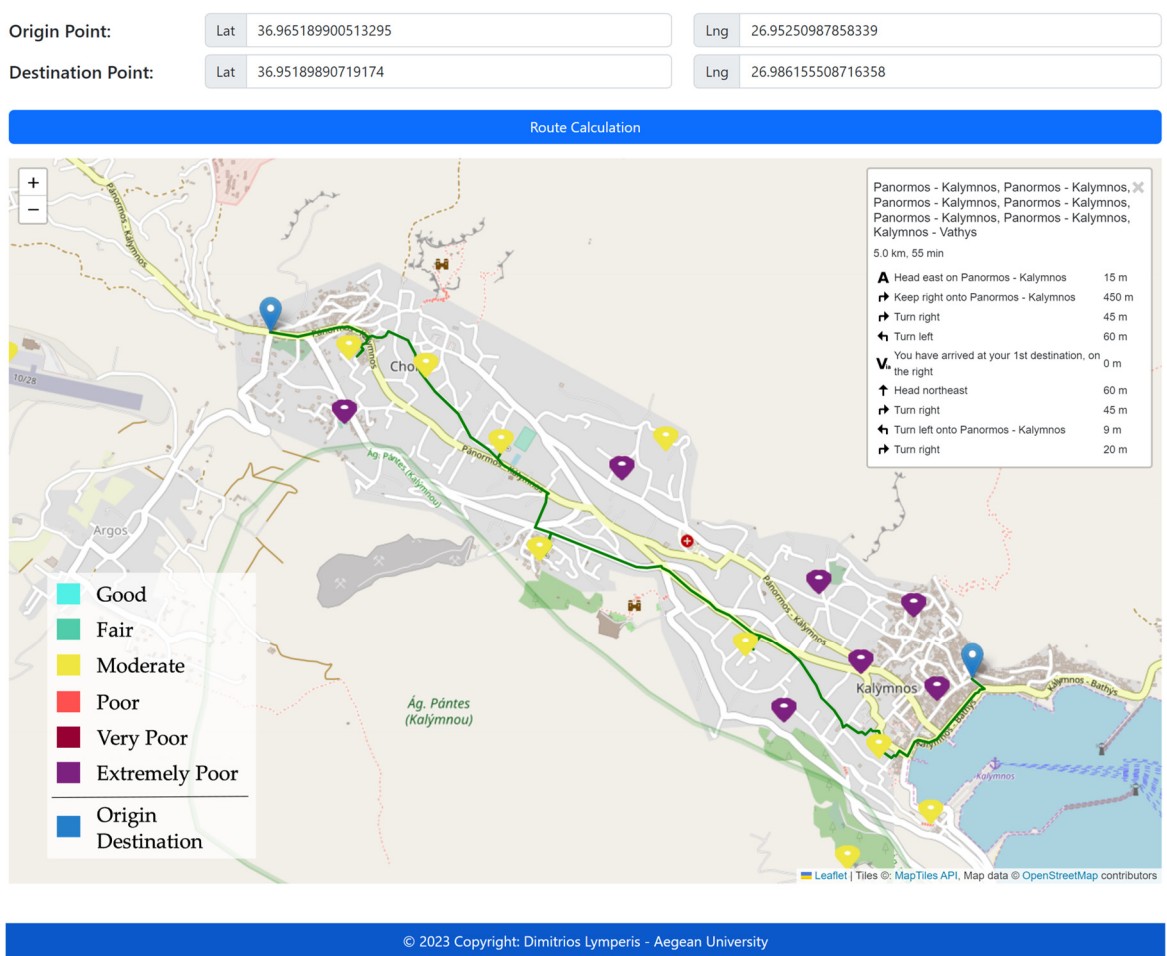

**Figure 14.** Illustration of a proposed "Green Route" on the map.

## 4. Performance Evaluation

Smart cities face the daunting challenge of managing and processing massive volumes of data produced by IoT devices and open-source platforms. As part of the evaluation of the SEDIA platform, three computational experiments were performed to evaluate the performance and scalability of the Semantic Layer as well as the retrieval of semantic data from the database, after a large number of concurrent requests. The experiments were conducted considering the PoC scenario presented in this study.

In the present evaluation, the system operates within the context of a VMware virtualization environment and is powered by an AMD EPYC 7543 32-Core Processor, of which a subset of eight cores is specifically allocated to the virtual machine under consideration. An integral component of this setup is the hierarchical cache memory structure with 256 KB provisioned for L1 data and instruction caches, 4 MB for L2, and a significant 256 MB for L3 cache, each cache instance being replicated eight times, paralleling the number of allocated cores. Furthermore, the system is equipped with 16 GB of memory, reinforcing its potential to handle sizable applications and tasks. Memory access optimization across the CPU landscape is realized through a single Non-Uniform Memory Access (NUMA) node configuration.

Within this virtual machine, three Docker containers, each with a discrete function, are deployed. The initial container hosts the "Green Route" Vue application. The second container utilizes Apollo Server to execute the GraphQL API. A Neo4j graph database resides in the third container. A virtual network facilitates the intercommunication and data exchange between these containers, ensuring their isolation and security.

Experiment1 aims to study the efficiency and scalability of the Service Layer in the proposed architecture, specifically focusing on the custom service processing MQTT messages from the IoT Middleware Layer. It measures total time spent on processing, annotating, and storing these messages in the Neo4j database to understand system performance and possible scalability issues. Experiment2 explores the performance of a Python service that uses asyncio for asynchronous operations to retrieve, process, and store data from open APIs into a Neo4j database. It specifically analyzes the processing time and impacts of asynchronous handling on the system's efficiency and effectiveness. Experiment3 is designed to test the system's performance under stress, by simultaneously simulating multiple clients retrieving environmental data from various IoT nodes. The objective is to understand the system's capability to handle real-world scenarios with a multitude of users concurrently fetching data. The sections that follow describe the methodology and outcomes of these experiments. For increased reliability and robustness of results, each experiment was conducted 20 times for each scenario investigated.

### 4.1. Experiment1: Performance Evaluation of MQTT Message Handling

The objective of this experiment is to thoroughly investigate the operational efficiency and scalability of the Service Layer in the proposed architectural design. Specifically, it evaluates the custom service responsible for processing MQTT messages received by the IoT Middleware Layer. This experiment focuses on recording the exact time requirements associated with processing received MQTT messages, semantically annotating them, and storing them in the Neo4j database. A comprehensive evaluation of this process allows for gaining an insightful understanding of system performance and potential scalability challenges.

This evaluation was achieved using MQTT JMeter plugin [46], a tool specifically designed for simulating MQTT protocol scenarios. To simulate varying loads of IoT devices publishing data, a script was created to generate multiple clients, each simulating a unique IoT device [47]. Each of these MQTT clients was set to publish messages to specific topics on the open source MQTT broker Mosquitto [48–50], emulating the behavior of a real-world IoT device sending data updates. The script was executed in Apache JMeter [51]. The published messages were then received, processed, and written into a Neo4j database.

In the computational experiment assessing the efficiency of an MQTT-based message processing system, the high-performance results are presented in Table 6. The minimum processing times showed just a slight decrease between 10 to 15,000 messages. This slight decline, albeit marginal, underlines the system's ability to maintain its efficiency under increased load. One observation regarding the maximum processing times is the system's performance when it processes larger quantities of messages, particularly when handling more than 8000 messages. Even at the upper limit of 2.0881 s for 15,000 messages, the system demonstrates exceptional adaptability to the sheer volume of incoming data.

**Table 6.** Experiment1 results: time metrics for processing MQTT messages across different volumes (measured in seconds).

| MQTT Messages | Min | Max | Avg | Std.dev |
|:---:|:---:|:---:|:---:|:---:|
| 10 | 0.0043 | 0.0056 | 0.0049 | 0.0005 |
| 100 | 0.0034 | 0.0107 | 0.0056 | 0.0012 |
| 1000 | 0.0034 | 0.0265 | 0.0055 | 0.0030 |
| 5000 | 0.0034 | 0.0399 | 0.0056 | 0.0033 |
| 8000 | 0.0034 | 2.0453 | 0.0058 | 0.0230 |
| 10,000 | 0.0033 | 2.0643 | 0.0058 | 0.0208 |
| 15,000 | 0.0033 | 2.0881 | 0.0057 | 0.0173 |

When considering the average processing times, they consistently remained low irrespective of the message quantity, further demonstrating the system's ability to maintain efficient message handling operations. The standard deviation values, which quantify the variation in processing times, progressively increase with the number of messages but

remain relatively low overall. This indicates a high level of consistency in processing times, ensuring reliable performance.

A critical component of this process, vital to achieving these time metrics, is the utilization of a *ThreadPoolExecutor*. Firstly, it allowed the MQTT clients to concurrently operate, thereby efficiently utilizing the system's resources. Each client, or thread, within the ThreadPoolExecutor's pool could independently process its incoming messages without blocking the execution of the other threads. This functionality is particularly vital when dealing with high-volume, real-time data, such as that from IoT devices, where delays in processing can result in data loss or inaccurate real-time analysis. Secondly, the ThreadPoolExecutor's internal management of the thread life cycle relieved the experiment from the complex task of manually creating, synchronizing, and terminating threads. By automatically managing a pool of worker threads, the ThreadPoolExecutor was able to swiftly allocate available threads to incoming tasks; i.e., processing incoming MQTT messages. Finally, in the context of the collected time metrics, the ThreadPoolExecutor allowed for the accurate measurement of message processing times within each thread. As each thread independently processed its messages, the processing time was not influenced by the concurrent execution of other threads. This aspect allowed the experiment to gather reliable and isolated timing metrics for each individual processing task.

In summary, the findings from this investigation attest to the system's scalability, robustness, and consistency. These favorable characteristics are critical in facilitating real-time data processing and subsequent database operations, thereby maximizing the efficiency of IoT-based systems and applications. The system's ability to maintain high performance while handling a considerable volume of messages makes it a promising tool for handling large data streams, a feature becoming increasingly crucial in the rapidly evolving IoT landscape.

### 4.2. Experiment2: Performance Evaluation of High Volume HTTP Requests

The second experiment aims to investigate the performance characteristics of a Python-based service that retrieves data from open sources, processes them, and stores them into the Neo4j database, using *asyncio* library for asynchronous operations. The experiment's focus is on measuring and analyzing the total time required for processing, annotating, and storing data from varying numbers of open data source API responses. The main goal of this experiment is to understand the impact of asynchronous processing on the system's efficiency and effectiveness when handling and writing responses into the Neo4j database.

In the experimental setup, a high-performance mocking server was employed, designed in Python using the FastAPI framework [52,53], to simulate high-throughput data responses. The server was capable of handling substantial concurrent request volumes, thus simulating scenarios of extreme load. It asynchronously operates, meaning it can concurrently manage multiple requests without impeding the execution of subsequent requests, thereby significantly enhancing its capacity to process a high volume of requests. Each GET request was responded to with a predetermined JSON payload, mimicking real air pollution data, thus accurately reflecting real open sources. The utilization of a mock server provided a controlled environment and consistent response data for stress testing, leading to reliable performance metrics.

In the assessment of the system's performance, various request-response scenarios were evaluated. In Table 7, the time taken for processing, formatting, annotating, and writing data into the Neo4j database is presented. The measurements do not include the time required for sending requests and receiving responses from the mocking server. Performance metrics for a system were gathered for request volumes ranging from 1000 to 40,000. Various statistical parameters, including minimum, maximum, average, 95th percentile, and standard deviation were measured to assess the system's responsiveness. This was determined by the elapsed time in seconds between sending a request and receiving a response.

**Table 7.** Experiment2 results: time metrics for asynchronous data retrieval and storage in Neo4j database (measured in seconds).

| Responses | Min | Max | Avg | 95th Pct | Std.dev |
|---|---|---|---|---|---|
| 1000 | 0.0200 | 2.0742 | 0.0230 | 0.0225 | 0.0649 |
| 5000 | 0.0200 | 2.1069 | 0.0212 | 0.0220 | 0.0295 |
| 10,000 | 0.0195 | 2.0978 | 0.0209 | 0.0219 | 0.0208 |
| 15,000 | 0.0184 | 2.0794 | 0.0202 | 0.0221 | 0.0169 |
| 20,000 | 0.0183 | 2.0939 | 0.0203 | 0.0223 | 0.0147 |
| 30,000 | 0.0185 | 2.0843 | 0.0210 | 0.0243 | 0.0120 |
| 40,000 | 0.0231 | 2.1471 | 0.0254 | 0.0298 | 0.0108 |

For the minimum response times, the system manifested minimal fluctuation, with values ranging from 0.0183 s to 0.0231 s. As the number of requests escalated from 1000 to 40,000, the system maintained a consistent minimum response time, indicating a robust capacity to handle requests in a timely manner, irrespective of the load. The maximum response times demonstrated relative stability, with values marginally varying from 2.0742 s to 2.1471 s. This suggests that even under heavy load, the system maintained its efficacy in processing, annotating, and storing the received data.

The consistency of the average response time indicates a consistent level of performance across a range of response numbers. It fluctuated between 0.0202 s and 0.0254 s, indicating the system's stability under varying load conditions. The 95th percentile response times, a crucial metric providing insights about the system's behavior under peak load, showed a slight increase as the number of requests escalated, although the increment was relatively moderate, ranging from 0.0220 s to 0.0298 s. This reflects the system's ability to maintain relatively swift processing, annotating, and storing for the majority of responses, even when under higher loads. The standard deviation progressively reduced as the number of requests increased. This reduction indicates an improvement in consistency, leading to a predictably steady response time under varying loads.

Utilizing asyncio, a crucial component of this process, is important for attaining these time metrics due to its core advantage of facilitating concurrent execution without the necessity of multithreading or multiprocessing. The execution flow, managed using coroutines, allows tasks to voluntarily relinquish control during I/O operations, permitting other tasks to execute, a key characteristic of event-driven programming libraries such as asyncio. This non-blocking approach to I/O operations can lead to significant performance improvements, particularly in situations where the system needs to handle a large number of simultaneous requests, as was the case in the scenarios tested. In the particular case under discussion, the system was dealing with many network or database I/O operations. The system was able to concurrently handle many requests, thus effectively utilizing resources and delivering high performance, as reflected by the low average and 95th percentile response times.

*4.3. Experiment3: Performance Analysis of GraphQL Query Execution on Neo4j*

Experiment3 aimed to assess the performance of semantic data retrieval under stress, by designing and executing a test that simulated multiple clients concurrently accessing environmental data from various IoT nodes. The synthetic workload was carefully crafted to imitate real-world situations where a multitude of users might use the application to simultaneously fetch data.

The experiment was conducted using Apache JMeter [51], an open-source software designed to load test functional behavior and measure performance. JMeter is particularly suited to this experiment due to its ability to simulate a heavy load on a server, group of servers, network, or object to test its strength or analyze overall performance under different load types. In this context, JMeter's ability to simulate many different users with concurrent GraphQL queries was exploited to create a robust and realistic testing environment.

Table 8 provides a comprehensive insight into the performance of a GraphQL server interfacing a Neo4j database, both running in separate Docker containers within a virtual network. The parameters evaluated include response times, failure rates, throughput, and network bandwidth consumption under varying user load conditions.

**Table 8.** Experiment3 results: performance metrics obtained from a GraphQL query accessing the Neo4j Database under varying user loads.

| GraphQL Query for Environmental Data Retrieval | | | | | | | | |
|---|---|---|---|---|---|---|---|---|
| Executions | | | Response Time (ms) | | | Throughput | Network (KB/s) | |
| # Users | FAIL | Error % | Min | Max | Average | Transactions/s | Received | Sent |
| 10 | 0 | 0.00% | 114 | 558 | 160 | 1.11 | 0.36 | 0.38 |
| 100 | 0 | 0.00% | 108 | 453 | 125 | 10.13 | 3.30 | 3.48 |
| 200 | 0 | 0.00% | 107 | 514 | 128 | 20.19 | 6.59 | 6.94 |
| 250 | 0 | 0.00% | 108 | 543 | 134 | 25.33 | 8.26 | 8.71 |
| 300 | 0 | 0.00% | 106 | 565 | 137 | 30.71 | 10.02 | 10.56 |
| 400 | 0 | 0.00% | 106 | 700 | 144 | 40.63 | 13.25 | 13.97 |
| 500 | 0 | 0.00% | 106 | 759 | 157 | 51.11 | 16.67 | 17.57 |
| 1000 | 0 | 0.00% | 105 | 1058 | 200 | 102.29 | 33.36 | 35.16 |
| 2000 | 0 | 0.00% | 1484 | 4832 | 2879 | 175.70 | 57.31 | 60.40 |
| 3000 | 57 | 1.90% | 1600 | 107,198 | 24,471 | 26.05 | 9.90 | 8.79 |
| 4000 | 623 | 15.57% | 4129 | 95,000 | 36,884 | 40.64 | 31.25 | 11.79 |

# It designates the number of users.

The performance metrics were observed for user loads ranging from 10 to 4000. For lower user loads up to 1000, the system demonstrated exceptional resilience and efficiency. The average response time maintained was under 200 ms. The system also showed a steady increase in throughput as the user load increased, achieving a peak of approximately 102 transactions per second at 1000 users. This high throughput indicates an efficient use of resources and well-optimized server and database operations.

A key aspect of system robustness, the failure rate, remained at 0% up to 2000 users, demonstrating the system's ability to handle substantial load without compromising the quality of service. The network metrics, both data sent and received, followed a linear trend, proportionally increasing with the user load, suggesting stable network performance.

However, beyond 1000 users, the system showed signs of strain. The average response times dramatically increased, peaking at over 36 s at 4000 users. Concurrently, the failure rate also began increasing beyond 2000 users, reaching up to 15.57% at 4000 users.

Despite these constraints under high load, it is critical to appreciate the overall performance of the GraphQL server and Neo4j setup in individual Docker containers. The system performance and scalability up to 1000 users are commendable, indicating well-optimized operations and effective resource utilization within each Docker container.

In conclusion, the successful management of up to 1000 simultaneous users with no transaction failures, low response times, and a stable network performance showcases the potential of the adopted architecture. The use of Docker containers allows for process isolation, straightforward setup, reproducibility, and the efficient use of resources, delivering a high-performance GraphQL server and Neo4j database system for a substantial user load. These attributes make this architectural approach a promising candidate for developing scalable, efficient, and reliable data retrieval systems.

## 5. Related Work

This section briefly presents and discusses some relevant studies that propose the use of semantic technologies for managing and interpreting sensor data, particularly in the context of environmental monitoring and smart cities. Most of the papers describe specific frameworks or systems that implement these technologies. The papers generally address challenges such as sensor heterogeneity, data quality, real-time processing, and scalability.

A semantic data model for the interpretation of environmental streaming data has been proposed in a relevant study [54]. The model uses a lightweight ontology approach to represent IoT data with the Semantic Web, using cross-domain knowledge to map sensor streaming data and annotation sensors. The proposed data model has several requirements, including data annotation, sensor and station information, lightweight data model, reuse domain knowledge, and semantic reasoning. The authors also discussed possible technological solutions, such as JSON-LD and RDF-Stream, and presented a conceptual data model. They then described the implementation strategy and prototype of their proposed model, which includes a set of URIs to identify sensors and sensor data, and an output of multi-data models that enables service and application.

A system architecture and implementation of an IoT real-time air quality monitoring system was presented in [55], where semantic annotations were integrated into sensor stream data for improved interpretation and understanding. The system receives raw sensor stream data in JSON format from the AQI API and processes it by integrating semantic annotations based on an ontology. The processed data is then displayed to users in real-time using an ASP.NET Core Model-View-Controller (MVC) application, Leaflet, and Apache Cassandra database. The authors also described the technologies and standards used in the proposed system, such as Spark Streaming, Apache Kafka, and Open Geospatial Consortium (OGC) standards. Overall, the system presented in this paper demonstrates the potential of integrating semantic annotations into sensor stream data for better understanding and decision-making in IoT applications, such as air quality monitoring.

An integrated system for real-time semantic annotation and interpretation of IoT sensor stream data, known as IoTSAS, has been proposed [56]. The system architecture comprises two main components: Real-Time Semantic Annotation (RTSA) and Real-Time Interpreting Semantically Annotated (RTISA). The system includes six modules: a real-time processing of integration and interpretation of semantics into sensor stream data module, a data modeling module, an IoT management metadata module, weather alerts and air quality monitoring modules, and an APIs for external systems module. To test the performance of the IoTSAS system, a sensor stream data simulator was developed, which generates pseudo-random sensor stream data by using the Random C# class in certain ranges defined for each parameter in the metadata module. The IoTSAS system was tested on five testing phases: unit test, integration test, system test, acceptance test, and performance testing. Based on the performance testing results, the IoTSAS system processed real time by annotating with semantics and interpreting the semantic annotations only for 138 s for 1,000,000 sensor observation data, proving the validity of high system performance.

Challenges in achieving interoperability among IoT devices and a proposed solution leveraging ontologies and machine learning were the focus of another relevant study [57]. The authors argued that the heterogeneity of IoT devices and the lack of a standard communication protocol between them make it difficult to share data and extract useful insights from it. To address this issue, they proposed a set of ontologies that can represent a high-level model of the IoT domain and enable interoperability between connected IoT systems. The authors used air quality monitoring as a use case to demonstrate how their proposed approach could work. They proposed a distributed architecture for processing optimization and collective performance, where a semantic gateway is used for inter-IoT communication, and two servers are used for data storage, preprocessing, and ontology. The proposed device model includes features such as time features, location, sensors, and metrics features. The authors suggested that machine learning models, such as Random Forest, could be used to extract more insights from the data provided by a single device. The ontology allows the researchers to gain better insight into the data communicated by the device, including Air Quality Index insights, as well as providing other insights about the location, features, and prediction metrics.

An IoT-based platform for environmental data sharing in smart cities, designed according to a three-layered IoT architecture, was discussed in [58]. The platform enables

the collection, storage, and processing of data from the city environment at a local region level, using Fog resources for local processing, and at the city level through services and resources dynamically deployed in the cloud. The platform uses the concept of adapters for the seamless integration of heterogeneous sensors and provides functionalities for Fog and Cloud interfaces to improve flexibility. The Sensor Markup Language (SenML) data representation format was adopted for compatibility, but the mapping of the Internationalized Resource Identifier (IRI) of the sensor is mandatory to reduce message overload and resource consumption in constrained devices. The platform enables the application of big data techniques and machine learning on city environmental data and defines the interrelation of data in a standardized and robust way. The manuscript also proposed the Environment Indicators Smart City Ontology (EISCO), which covers a broad set of environmental indicators, making possible the definition of data semantics for Linked Data Storage, and enabling the extraction of multidomain knowledge.

A Semantic Smart World Framework (SSWF) for constructing a general semantic big data framework for a smart world was the subject of the work discussed in [59]. The framework includes a universal knowledge base, association rule discovery, Service-Oriented Architecture (SOA), and the use of semantic RDF standards. The paper presented a case study of the SSWF to analyze air pollution and weather on migratory birds' paths, and the results showed high accuracy in prediction and matching with real data. This study also revealed a correlation between escalating air pollution levels and changing weather conditions. Experimental results suggest that the framework offers satisfactory performance in handling heterogeneous big data.

Another semantic framework that integrates IoT with machine learning techniques for smart cities was proposed in [60]. The framework leverages an urban knowledge graph to model and reason regarding city data, and it is designed to handle heterogeneous and dynamic data streams. The authors presented two use cases to demonstrate the effectiveness of the proposed framework: pollution detection from vehicles and traffic pattern analysis. The experiments showed that the framework is scalable and efficient, and it can provide useful insights for decision-making in smart cities. However, the study also has some limitations, such as missing data and limited availability of open data, which the authors plan to address in future work.

Challenges and solutions associated with implementing a semantic data management system for air quality monitoring in smart cities are the main focus of the OpenSense project [61]. This project aims to provide transparent access to air quality data by generating meaningful and semantically understandable data. The paper describes a layered semantic data management model that adds value to the data through semantic annotations that describe data cleaning and preprocessing, temporal and spatial aggregations, and event annotations. The authors believe that this type of deployment can form the basis for a city-wide infrastructure that enables self-monitoring and self-healing using semantic data management tools and technologies.

The proposed approach in this study stands out from related works as it provides a comprehensive platform that covers all stages of data management, from gathering heterogeneous data to semantic labeling and storage, and from analyzing the data to presenting the extracted knowledge to users through web applications. Unlike previous studies, this methodology dynamically creates relationships based on the geographical distance between entities, resulting in more accurate and relevant results. Additionally, the system utilizes open data sources and incorporates them into the analysis process, further enriching the collected data. Overall, this methodology provides a more complete and effective approach to data collection and analysis, making it a valuable contribution to the field of geospatial data management.

Table 9 presents a qualitative comparison of the presented semantic integration platforms for smart city applications, including the approach proposed in this study. The platforms are evaluated based on their semantic data models, annotation capabilities, rea-

soning, real-time processing, dynamic relationships, geographical data handling, machine learning use, data sources, PoC application, and performance evaluation.

**Table 9.** Qualitative comparison of semantic integration platforms for Smart City Applications.

| Platform | Semantic Data Model [1] | Semantic Annotation | Semantic Reasoning | Real Time Processing | Dynamic Relationships | Geospatial Data | Machine Learning | Data Sources | PoC [2] | Perf. Eval. [3] | Ref. |
|---|---|---|---|---|---|---|---|---|---|---|---|
| Duy et al. | Semantic Ontology (SSN, TIME, SWEET, GEO) | ✔ | ✔ | ✖ | ✖ | ✔ | ✖ | IoT | ✖ | None | [54] |
| Sejdiu et al. | Onto-Core (SensorML, O&M, Tans-duserML) | ✔ | ✖ | ✔ | ✖ | ✔ | ✖ | IoT | ✖ | None | [55] |
| IoTSAS | Onto-Core (SensorML, Tans-duserML, O&M, SOS, WNS, SAS, SPS) | ✔ | ✖ | ✔ | ✖ | ✔ | ✖ | IoT | ✔ | Limited | [56] |
| Noussair et al. | DeviceModel (Geonames, DBpedia) | ✖ | ✖ | ✖ | ✖ | ✔ | Random Forests | IoT | ✖ | None | [57] |
| Rubi et al. | EISCO (ENVO, GCIO, SSN) | ✖ | ✖ | ✖ | ✖ | ✔ | ✖ | IoT, Open Data | ✖ | Limited | [58] |
| SSFW | SCEO (GEO W3C) | ✖ | ✔ | ✖ | ✖ | ✔ | ✖ | IoT, social media | ✖ | Limited | [59] |
| Zhang et al. | Urban Knowledge Graph | ✖ | ✖ | ✖ | ✖ | ✔ | Transfer Learning | IoT, multimedia content, social media, and crowd-sourcing | ✖ | Limited | [60] |
| OpenSense | OpenSense (SSN) | ✔ | ✖ | ✔ | ✖ | ✔ | ✖ | IoT crowd-sourcing | ✔ | None | [61] |
| SEDIA | SEDIA (SSN, iot-lite, GeoSPARQL) | ✔ | ✔ | ✔ | ✔ [4] | ✔ | ✖ | IoT, Open Data | ✔ | Extensive | this work |

[1]: Semantic models that were employed in the relevant study are presented in this column. Where the suggested ontology is specified, its name is given, followed by the collection of semantic standards or ontologies that were employed in parentheses. [2]: The PoC column refers to the comprehensive, detailed account of the PoC, going beyond a mere overview to provide an in-depth understanding of its functionalities and operations. [3]: There are three distinct levels for the evaluation of performance: 'Extensive' which involves rigorous and comprehensive testing to evaluate performance across multiple parameters; 'Limited' which focuses on testing only select features or under specific conditions; and 'None' where no testing is performed. [4]: Based on the geographical distance between entities.

## 6. Discussion and Future Directions

The rapidly expanding landscape of IoT technology, an integral component of smart city development, brings forth an unprecedented proliferation of data, marked not only by its immense volume but also by its striking heterogeneity. As IoT devices, each equipped with unique sensors and interfaces, continuously generate a wealth of data, they consequently contribute to a flood of information that is as vast as it is diverse. This vast universe of information is marked by significant format and spatial heterogeneity. Format heterogeneity arises from the multitude of data types and formats generated by varying IoT devices. Each device, aligned with its specific functionality and design, produces data that are fundamentally different in structure and form. Spatial heterogeneity, in contrast, emerges from the extensive geographical distribution of IoT devices. Data produced reflects a broad spectrum of local conditions, underscoring the need for sophisticated analytics capable of comprehending and integrating these diverse datasets. This dual heterogeneity, while presenting both an opportunity for garnering rich insights and a barrier to seamless data integration, poses a significant challenge to the effective operation and evolution of smart cities.

The SEDIA architecture proposed in this paper offers a promising platform for integrating diverse geographical data that may be utilized across a wide variety of application areas. The architecture's components are capable of handling diverse data sources, including geographical information, and support a semantically enriched data model that facilitates effective data integration and analysis. The implementation of SEDIA on top of an existing

IoT middleware enhances its services, capitalizes on abstraction levels, and fosters interoperability. The PoC smart city application related to air quality monitoring demonstrates the efficacy of SEDIA in identifying patterns and relationships within the data.

Three computational experiments were performed to evaluate the efficiency and the potential scalability of the proposed system architecture. The first scrutinized the Service Layer's effectiveness in processing MQTT messages from the Middleware Layer. The overall results of this experiment underscore the system's scalability, robustness, and consistency, vital for maximizing the efficiency of IoT-based systems and applications. The demonstrated high-performance handling of substantial message volume, backed by the effective use of a ThreadPoolExecutor, bolsters the system's potential as a tool for managing large data streams in the rapidly expanding IoT landscape.

The second experiment examined the performance of the Service Layer's effectiveness in processing data from open sources through APIs. The experiment demonstrated the system's robust capacity and reliability in managing a significant volume of requests, exhibiting consistently low response times. The utilization of asyncio library played a crucial role in achieving these metrics, as its non-blocking approach facilitated the efficient handling of numerous simultaneous requests. Further research could explore potential enhancements to maximize response times and reduce variability in response times. Nonetheless, the current findings reinforce the system's prowess in managing substantial request volumes while maintaining consistently satisfactory performance.

The third experiment simulated a scenario of stress performance of semantic data retrieval from the Semantic Layer with concurrent users accessing environmental data from various IoT nodes. Despite the constraints under high load, the system's performance and scalability of up to 1000 users were laudable, demonstrating the effectiveness of the Docker-contained GraphQL server and Neo4j setup.

Further optimization could focus on handling higher user loads to enhance scalability and resilience. In the current research, a virtual environment where three Docker containers are used, each with a distinct purpose, was considered. The first container runs a "Green Route" Vue application, offering hosting services. The second container uses an Apollo server to run the GraphQL API, and the third container hosts a Neo4j graph database. These containers are connected to each other through a virtual network that facilitates smooth communication and secure data exchange while maintaining their isolation and security. For future work, it is suggested to integrate this Docker installation with Kubernetes, load balancing and replica sets to improve system scalability and resilience, ensuring high availability under variable load conditions. This integration has the potential to significantly improve the platforms' ability to handle increased workloads while maintaining high performance and uptime. Kubernetes can automate the deployment, scaling, and management of containerized applications. Applying Kubernetes to our existing system can ensure the seamless operation of Docker containers while automating scaling operations based on workload and performance metrics, enabling more efficient use of resources [62–64].

In addition, by incorporating load balancing, which refers to evenly distributing network traffic across multiple servers, this approach can avoid overloading a single server. This strategy enhances system resilience by helping to evenly distribute work across containers, thereby increasing overall system performance. Finally, it is intended to explore the use of ReplicaSets in Kubernetes, which ensures that a certain number of identical pod versions are running at any time [65]. This approach protects system availability in the event that a pod or host machine fails. By including multiple copies of each application component in the Kubernetes configuration, improvements in system availability, concurrent user handling, and disaster recovery capabilities are expected. While such integration introduces additional complexity to initial installation and ongoing maintenance, the expected improvements in user experience and system performance could provide significant rewards.

Undoubtedly, the incorporation of security protocols into the communication process introduces additional computational costs, thereby contributing to an increase in overall

processing time overhead. The term 'overhead' may refer to multiple aspects, including additional packet transfers, increased latency, and the system's ability to efficiently scale. Our discussion on the overhead of secure data transmission is focused on the utilization of MQTT with TLS, which was the main mechanism employed in our work. The TLS version we adopted in our work is TLS 1.3, which has been made more secure and efficient by removing obsolete and weak security algorithms, and reducing the number of exchanges needed to complete the handshake. According to a recent study, implementing security measures such as MQTT with TLS can contribute a significant overhead, leading to a 10–20% increase in mean response time when deploying brokers and clients across a cloud platform and exchanging MQTT messages using QoS level 0 on a variety of topics [66]. In this scenario, the mean response time ranged from 2 to 12 ms. In another study, the overheads in IoT systems were examined by comparing the more reliable MQTT and the less reliable CoAP on an Arduino Uno test-bed. The findings indicate that while MQTT provides a more reliable infrastructure, it incurs higher power consumption and latency compared to CoAP [67].

In certain scenarios, specifically in hard real-time applications such as control processes for smart grids and microgrids (where the maximum permissible response time can be as low as 100 ms), the overhead imposed by the MQTT + TLS scheme may render it less than ideal [68]. In an IoT setup, akin to the one employed in this study, which utilized multiple Arduino Uno devices with WiFi and MQTT + TLS at QoS level 0, the measured processing time for secure transactions of 5 KB (21 packets) indicated a latency of 140 ms. In contrast, the latency was a mere 7 ms when security mechanisms were not implemented. Other schemes such as CoAP + TLS may be more suitable for such stringent latency requirements. On the other hand, the scalability of the MQTT + TLS scheme, defined as the change in total response time for a control process as the number of tasks increases, has proven to be linear compared to the CoAP + TLS scheme which was found to be exponential [68]. This is largely because the TLS security channel in MQTT needs to be established only once at the beginning, leading to a minimal impact on the response time as the number of tasks increases. Given the similarities in the IoT devices and security mechanisms used in our work it is reasonable to assert that despite the inherent overhead, the MQTT + TLS scheme provides a suitable choice for smart city applications, particularly for soft real-time scenarios such as the green route example discussed in this paper.

As the present research is focused on scalability in a containerized environment, the performance implications of operating Docker containers in a virtual machine (VM) environment must be considered. While containerization offers several benefits such as portability and ease of deployment, it is essential to consider the impact it may have on system latency, especially when dealing with cloud-to-edge communications. To assess VM-based containerization overhead a simple experiment was conducted. In this experiment, network latency was measured using Netperf's request/response mode [69]. The client runs on one host while the server runs on another host, both with Docker and without Docker for comparison. The client sends 100-byte messages to the server. Upon receipt of each message, the server immediately responds, creating a round trip. This process continuously repeats, and the transaction rate, i.e., the number of request/response pairs completed per second, allows for the calculation of network latency. The findings from this experiment indicate that the latency of a Docker container running in a VM (0.093 ms) is 2.5 times higher compared to a non-containerized server (0.037 ms). This is primarily due to Docker's use of Network Address Translation (NAT) to manage network ports and the added virtual device layer created by the VM [70]. Both of these factors result in extra processing steps, which contribute to increased latency. Future work would benefit from evaluating the system's performance by operating containers directly on the bare metal to avoid virtualization overheads and leveraging modern container technologies such as Kubernetes.

The performance implications of mapping cloud services to edge nodes require consideration of several crucial factors, including transmission time influenced by distance,

transmission medium, and network traffic, the processing time for data preparation and server processing, propagation delay, queuing time due to network congestion, encryption and decryption times for security protocols, communication protocol overhead, container-related delays, network jitter, and hardware performance of IoT devices and servers. These factors collectively impact latency in the communication between cloud services and edge nodes. Detailed performance analysis of the SEDIA platform based on the above factors will be conducted in the future.

Additionally, as future work, machine learning algorithms can be incorporated into the Application Layer of SEDIA to further process and analyze the retrieved data. This can enable the generation of valuable insights, predictions, and recommendations that can enhance the overall performance and effectiveness of the proposed solution. For example, machine learning algorithms can be used to predict air quality levels based on historical data and other environmental factors. Additionally, machine learning can be used to identify patterns and relationships within the data that may not be immediately apparent, enabling more effective decision-making and resource allocation. Therefore, incorporating machine learning algorithms into SEDIA can enhance the platform's capabilities and enable more effective monitoring and mitigation of various urban issues.

## 7. Conclusions

In this paper, we introduced SEDIA, a novel platform designed for data management and geographical information utilization, specifically tailored for the development of smart city applications. SEDIA stands out by providing a comprehensive solution that encompasses the full cycle of data handling: gathering, semantic labeling, storage, analysis, and presentation. SEDIA places emphasis on semantic enrichment and geographical relationships, which are not typically at the forefront of similar studies. The platform utilizes ontology classes and properties to semantically annotate collected data, and the Neo4j graph database to facilitate the recognition of patterns and relationships within the data. The PoC smart city application related to air quality monitoring demonstrated the efficacy of SEDIA in monitoring and mitigating air pollution in urban environments. The implications of this research are significant, as SEDIA has the potential to be used in a wide range of smart city applications beyond air quality monitoring, ultimately improving the quality of life for citizens.

**Author Contributions:** Conceptualization, C.G.; Methodology, D.L. and C.G.; Software, D.L.; Validation, D.L. and C.G.; Formal analysis, C.G.; Investigation, D.L. and C.G.; Resources, D.L. and C.G.; Data curation, C.G.; Writing—original draft, D.L.; Writing—review & editing, C.G.; Visualization, D.L. and C.G.; Supervision, C.G.; Project administration, C.G.; Funding acquisition, C.G. All authors have read and agreed to the published version of the manuscript.

**Funding:** This research was funded by the "Research e-Infrastructure [e-Aegean R&D Network] Action 1.2 e-Aegean Geospatial data services" project, Code Number MIS 5046494, which is implemented within the framework of the "Regional Excellence" Action of the Operational Program "Competitiveness, Entrepreneurship and Innovation". The action was cofunded by the European Regional Development Fund (ERDF) and the Greek State [Partnership and Cooperation Agreement 2014–2020].

**Data Availability Statement:** Data will be made available on request.

**Acknowledgments:** We express our gratitude to our colleagues who have been involved in the "Research e-Infrastructure [e-Aegean R&D Network] Action 1.2 e-Aegean Geospatial data services" project for their valuable support throughout the research and development activities presented in this study.

**Conflicts of Interest:** The authors declare no conflict of interest.

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
