# Peer review of "SEDIA: A Platform for Semantically Enriched IoT Data Integration and Development of Smart City Applications"

_futureinternet, doi:10.3390/fi15080276_

Round 1

Reviewer 1 Report

The authors have proposed a framework SEDIA, a platform for developing smart applications based on diverse data sources, including geographical information, to support a semantically enriched data model for effective data analysis and integration. The paper need major revisions and my comments are as following. 

1. As the authors claim that SEDIA can handle heterogenous data and this can simplify the data integration aspect. This is not the case because of the following reasons. Authors should address this very carefully in the revised manuscript. 

1a. If the IoT devices are from multiple manufacturers, there are 100% chance that they will have diversified communication protocols. That is the case in real-time networks. Not all can communicate with MQTT. They can be DNP3, they can be IEC61850 or etc. How this setup will simplify this heterogeneous architecture. 

1b. Other issue is IoT middleware. Is it edge or fog device. Because placement of this device will pave the way of communication bandwidths in the network. This should be clarified. 

1c. Moreover, what is the overhead for secure data transmission. 

2. As the manuscript explains LoraWAN communications between the sensor nodes and main server, is there any work off-loading mechanism. Because under extreme conditions and data traffic, the system will not be able to communicate efficiently. It is nice to discuss task scheduling for each node. 

3. For cloud computing side, docker containers are used that may increase the latency of the overall system especially when you have to map any service from the cloud to the edge nodes. How that performance is calculated here. 

English is satisfactory. Few grammatical mistakes should be improved. 

Reviewer 2 Report

The goal of this paper, as exposed by the authors, is to introduces SEDIA, a platform for developing smart applications based on diverse data sources, including geographical information, to support a semantically enriched data model for effective data analysis and integration.

Please Quantify the main contribution in the introduction and conclusion sections, compared to other authors' articles. For each point mentioned in the contribution paragraph, authors need to identify which part in the submitted manuscript considers that point.

The paper present in detail the problems, current limitations and challenges of researchers regarding the smart cities, semantic enrichment, geospatial data in the context of IoT and air pollution.  

The reference section is good, citing new and relevant articles in the research area.
